# Genomics of soil depth niche partitioning in the Thaumarchaeota family Gagatemarchaeaceae

Paul O. Sheridan [1,2], Yiyu Meng[1], Tom A. Williams [3] &
Cécile Gubry-Rangin [1] ✉

Knowledge of deeply-rooted non-ammonia oxidising Thaumarchaeota lineages from terrestrial environments is scarce, despite their abundance in acidic soils. Here, 15 new deeply-rooted thaumarchaeotal genomes were assembled from acidic topsoils (0-15 cm) and subsoils (30-60 cm), corresponding to two genera of terrestrially prevalent Gagatemarchaeaceae (previously known as thaumarchaeotal Group I.1c) and to a novel genus of heterotrophic terrestrial Thaumarchaeota. Unlike previous predictions, metabolic annotations suggest Gagatemarchaeaceae perform aerobic respiration and use various organic carbon sources. Evolutionary divergence between topsoil and subsoil lineages happened early in Gagatemarchaeaceae history, with significant metabolic and genomic trait differences. Reconstruction of the evolutionary mechanisms showed that the genome expansion in topsoil Gagatemarchaeaceae resulted from extensive early lateral gene acquisition, followed by progressive gene duplication throughout evolutionary history. Ancestral trait reconstruction using the expanded genomic diversity also did not support the previous hypothesis of a thermophilic last common ancestor of the ammonia-oxidising archaea. Ultimately, this study provides a good model for studying mechanisms driving niche partitioning between spatially related ecosystems.

Many microbial genomes have been released recently due to the advent of culture-independent whole-genome sequencing techniques, including genome-resolved metagenomics. Concurrently, recently developed phylogenomic approaches such as gene tree - species tree reconciliation have enabled the investigation of mechanisms of genome evolution across large evolutionary timescales. These approaches have been applied to understand microbial habitat transitions between different ecosystems[1,2], such as from aquatic to terrestrial environments, and dramatic niche transitions between, for example, free-living and host-associated lifestyles[3–5]. However, the adaptive mechanisms associated with ancestral niche specialisation between spatially closely related ecosystems, such as associated topsoils and subsoils, have not been investigated.

Thaumarchaeota are commonly known for their ammonia oxidation function, which is a crucial step in the global nitrogen cycle[6]. However, this metabolism appears restricted to a single class within this phylum (Nitrososphaeria), with deeply-rooted Thaumarchaeota lacking potential for ammonia oxidation in soil[7–10], hot springs[11,12] or marine environments[13,14]. Instead, these non-ammonia oxidising archaea (non-AOA) Thaumarchaeota produce energy using sulphur and iron-reduction[11,12] or utilisation of organic substrates[13–15]. This Thaumarchaeota diversity offers the opportunity to address open questions in the evolution of the phylum, such as speciation in diverse environments.

The deeply-rooted Group I.1c Thaumarchaeota[10] are prevalent in terrestrial ecosystems, particularly in forest soils where they can comprise 20-25% of the archaeal abundance[9]. Their role in soil ecology

[1]School of Biological Sciences, University of Aberdeen, Aberdeen, UK. [2]School of Biological and Chemical Sciences, University of Galway, Galway, Ireland. [3]School of Biological Sciences, University of Bristol, Bristol, UK. ✉e-mail: c.rangin@abdn.ac.uk

is largely unknown, but an analysis of a single representative genome, Fn1, suggested that they are anaerobic heterotrophs[15]. However, this prediction contradicts the observed aerobic growth of Group I.1c Thaumarchaeota in soil microcosms[16]. Group I.1c Thaumarchaeota are present in both topsoils and subsoils[16,17], with distinct lineages being differentially abundant at different soil depths[16]. This depth-based niche partitioning provides a strong model for studying the ecological and evolutionary niche specialisation to soil depth in archaea.

Following metagenome assemblies from topsoils and subsoils, we assembled 15 new archaeal genomes and characterised Group I.1c Thaumarchaeota as a novel archaeal family (*Candidatus* Gagatemarchaeaceae). This family is prevalent in acidic soils and appears to have undergone an early evolutionary divergence, with distinct lineages occupying topsoils and subsoils. The early split between both lineages corresponds with significant genomic differences and specialised metabolisms. A gene tree-species tree reconciliation approach revealed that the early acquisition of novel gene families, followed by extensive gene duplication, drove the genome diversification of these archaea.

## Results

### Assembly and classification of non-ammonia oxidising Thaumarchaeota genomes

Fifteen Thaumarchaeota metagenome-assembled genomes (MAGs) that represent novel species of terrestrial archaea based on GTDB relative evolutionary divergence scores were recovered from five topsoils (0–15 cm) and four subsoils (30–60 cm), all acidic (Table 1). These genomes were related to the non-AOA Thaumarchaeota. Thirteen of the new genomes were affiliated with the Group I.1c clade (represented as f_UBA183 in GTDB). Two genomes were classified as members of the uncharacterised f_UBA141 family, a family closely related to the heterotrophic marine Thaumarchaeota (HMT)[13,14] (classified as f_UBA57 in GTDB). The ammonia monooxygenase *amoA* or *amoB* genes were not detected in any of the 15 genomes using BLASTn[18] or BLASTp against custom databases of *amoA* and *amoB* sequences[19], by GhostKOALA[20], or by hmmsearch[21] (*amoA*; PF12942, *amoB*; PF04744) indicating that these organisms are likely not capable of ammonia oxidation. The newly assembled Thaumarchaeota genomes were of relatively high quality, with average completeness of 70% (range: 49–95%) and average contamination of 2% (range: 0–9%) (Table 1). These genomes were predicted to be at relatively low abundance within their environments, averaging 0.7% (range: 0.1–3.1%) based on metagenomics sequence read recruitment (Table 1, Supplementary Data 1).

### Diversity and prevalence of Group I.1c

The 13 new Group I.1c genomes and the closely related publicly available genomes (Fn1, YP1-bin3, UBA183, palsa-1368, bog-1367 and bog-1369) belong to a single family and represent two genera and 17 species (Supplementary Data 1–3) according to the GTDB-Tk and AAI criteria outlined in the Methods section. The inferred phylogeny of Thaumarchaeota reveals a significant split between lineages occupying topsoils and subsoils, indicating that specialisation in these different habitats occurred early in their evolution (Fig. 1). Based on the current genomic representation, subsequent habitat switching does not appear to have happened since the divergence (Fig. 1). Using representative 16 S rRNA gene sequences from each of the two Group I.1c lineages, it was observed that the Group I.1c family was detected in diverse environments and is particularly prevalent in peat and cave soils (present in 44 and 30% of 16 S rRNA sequencing libraries, respectively) (Fig. 2A, Supplementary Data 4). Subsoil Group I.1c are twice as prevalent as topsoil Group I.1c in peat (11 versus 6%), whereas topsoil Group I.1c are 4-fold more prevalent than subsoil Group I.1c in more than 67,000 soils (7 versus 2%) (Fig. 2B, Supplementary Data 5).

Competitive read recruitment of metagenomic reads from the 15 soils (Supplementary Data 6) against Group I.1c genomes revealed that topsoil and subsoil Group I.1c lineages are differentially abundant in the two soil layers (*P* < 0.01) (Supplementary Data 7), indicating niche partitioning between these diverging lineages. The topsoil lineage dominates the Group I.1c community in topsoil soils, with the subsoil lineage comprising only 10% of the total Group I.1c abundance (Supplementary Fig. 1). The subsoil lineage makes up a significantly higher proportion of the Group I.1c community (41%; *P* < 0.01) at a depth of 30–45 cm (Supplementary Fig. 1) than in the topsoil environment. The proportion of subsoil lineage appears to increase further at depths of 45–60 cm (76%) (Supplementary Fig. 1).

Classification of the acquired Group I.1 c genomes against previously published phylogenetic groups of Group I.1c[19] indicates that most Group I.1c topsoil genomes belong to the terrestrial Group I.1c GC1 and GC5 groups (Supplementary Data 8) (Supplementary Fig. 2), which have been shown to grow under aerobic conditions[16]. In contrast, most Group I.1c subsoil genomes belong to the GC7 group (Supplementary Data 8), which was more abundant in subsoil than in topsoil forest soil previously studied[16].

With regards to formal taxonomic classification, we selected the genome bog-1369 as type material for classifying the novel family comprising Group I.1c (henceforth Gagatemarchaeaceae). Bog-1369 and Fn1 genomes were selected as type materials for classifying the novel topsoil (henceforth *Gagatemarchaeum*) and subsoil (henceforth *Subgagatemarchaeum*) genera, respectively. These genomes meet the quality criteria for type material suggested for MIMAGs[22,23], including high genome completeness ( > 95% complete) and possessing the 5 S, 16 S and 23 S rRNA genes (Supplementary Data 9). Full classification notes are detailed in Supplementary Note 1: Classifications.

### Shared metabolism within Gagatemarchaeaceae genomes

None of the Gagatemarchaeaceae genomes possessed the ammonia monooxygenase genes, suggesting that they cannot oxidise ammonia as an energy source (Supplementary Data 10). They lack marker genes of dicarboxylate-hydroxybutyrate, reductive acetyl-CoA and Wood-Ljungdahl carbon fixation pathways and also lack the hydroxypropionate-hydroxybutyrate pathway common in ammonia-oxidising archaea (AOA). Only three topsoil genomes possess the Type III ribulose-bisphosphate carboxylase (*rbcL*) and the ribose 1,5-bisphosphate isomerase (predicted to be involved in thaumarchaeotal RuBisCo[11]), indicating carbon fixation through the RuBisCo system (Supplementary Data 10). These two genes and the ribose-phosphate pyrophosphokinase were found to be adjacent to each other in these genomes, but all three genes were absent from other members of the family (Supplementary Data 11). Therefore, most Gagatemarchaeaceae likely acquire carbon from organic sources, such as exogenous carbohydrates, amino acids and fatty acids.

Gagatemarchaeaceae encode multiple genes involved in complex carbohydrate degradation, including glycoside hydrolases, carbohydrate esterases and auxiliary activity enzymes (Supplementary Data 12), as well as multiple GH135, GT39 and GH92 genes that possess signal peptides, indicating those that are secreted (Supplementary Data 13). Enzymes of the GH135 and GT39 CAZyme families are involved in the degradation and modification of fungal cell wall components[24,25], while GH92 enzymes cleave alpha-mannans (a major component of fungal cell walls)[26,27], potentially providing a carbon and nitrogen source for Gagatemarchaeaceae. The family lacks a complete glycolytic pathway, notably lacking key glycolytic gene phosphofructokinase, but could possibly metabolise carbohydrates through the pentose phosphate pathway.

Gagatemarchaeaceae also encode multiple genes involved in peptide degradation (Supplementary Data 14), including several signal peptide-encoding peptidases (Supplementary Data 15). These putatively extracellular enzymes include S01C and S09X family serine peptidases, which are also encoded by several members of the AOA (Supplementary Data 15), and the peptidases S53 and A05

**Table 1 | Genome characteristics of newly sequenced metagenome-assembled genomes**

| Short name | Completeness (%) | Contamination (%) | Relative abundance (%)* | Optimal growth temperature (°C)** | GC% | Adjusted genome size (bp) | Number Contigs | Adjusted CDS number | Environment source | Type of Soil | Soil pH | Soil Depth (cm) |
|---|---|---|---|---|---|---|---|---|---|---|---|---|
| *Ca. Gagatemarchaeum* | | | | | | | | | | | | |
| AcS1-13 | 70.9 | 0.0 | 0.75 | 37 | 62 | 2.3.E+06 | 85 | 2596 | Topsoil | Humus-iron podzols | 4.2 | 0–15 |
| AcS1-27 | 71.4 | 0.0 | 0.38 | 36 | 59 | 3.0.E+06 | 122 | 3234 | Topsoil | Humus-iron podzols | 4.2 | 0–15 |
| AcS1-6 | 85.0 | 5.8 | 0.81 | 38 | 59 | 2.4.E+06 | 52 | 2554 | Topsoil | Humus-iron podzols | 4.2 | 0–15 |
| AcS4-109 | 82.9 | 1.9 | 0.10 | 37 | 60 | 2.9.E+06 | 204 | 3124 | Topsoil | Humus-iron podzols | 4.9 | 0–15 |
| AcS5-19 | 89.8 | 2.6 | 0.14 | 34 | 58 | 2.3.E+06 | 154 | 2639 | Topsoil | Humus-iron podzols | 3.7 | 0–15 |
| AcS9-25 | 53.1 | 3.0 | 0.24 | 36 | 61 | 2.2.E+06 | 165 | 2782 | Topsoil | Peaty gleyed podzols | 4.4 | 0–15 |
| AcS11-71 | 54.2 | 9.5 | 0.20 | 37 | 60 | 3.4.E+06 | 1219 | 4973 | Topsoil | Humus-iron podzols | 4.0 | 0–15 |
| *Ca. Subgagatemarchaeum* | | | | | | | | | | | | |
| SubAcS9-116 | 95.3 | 1.0 | 0.40 | 39 | 59 | 1.7.E+06 | 26 | 1886 | Subsoil | Peaty gleyed podzols | 4.9 | 45–60 |
| SubAcS10-18 | 54.1 | 0.0 | 0.47 | 41 | 57 | 4.5.E+05 | 48 | 527 | Subsoil | Noncalcareous gley | 4.6 | 30–45 |
| SubAcS11-97 | 50.1 | 9.0 | 0.37 | 39 | 57 | 2.5.E+06 | 398 | 3039 | Subsoil | Humus-iron podzols | 3.9 | 30–45 |
| SubAcS15-15 | 60.7 | 1.0 | 2.35 | 37 | 54 | 2.0.E+06 | 3 | 2225 | Subsoil | Peaty gleyed podzols | 5.0 | 45–60 |
| SubAcS15-57 | 94.7 | 1.0 | 0.22 | 39 | 56 | 2.2.E+06 | 151 | 2426 | Subsoil | Peaty gleyed podzols | 5.0 | 45–60 |
| SubAcS15-94 | 48.9 | 0.0 | 3.09 | 41 | 57 | 1.4.E+06 | 71 | 1594 | Subsoil | Peaty gleyed podzols | 5.0 | 45–60 |
| *Heterotrophic Terrestrial Thaumarchaeota* | | | | | | | | | | | | |
| SubAcS9-71 | 74.3 | 1.0 | 0.35 | 33 | 46 | 1.8.E+06 | 214 | 2009 | Subsoil | Peaty gleyed podzols | 4.9 | 45–60 |
| SubAcS15-91 | 57.6 | 1.0 | 0.17 | 35 | 48 | 3.8.E+06 | 378 | 4151 | Subsoil | Peaty gleyed podzols | 5.0 | 45–60 |

Genome size and CDS number were adjusted for completeness. More detailed information on all genomes used in this study can be found in Supplementary Data 1. * Relative abundance was based on metagenomic read recruitment to genomes. **Optimal growth temperature was predicted in silico.

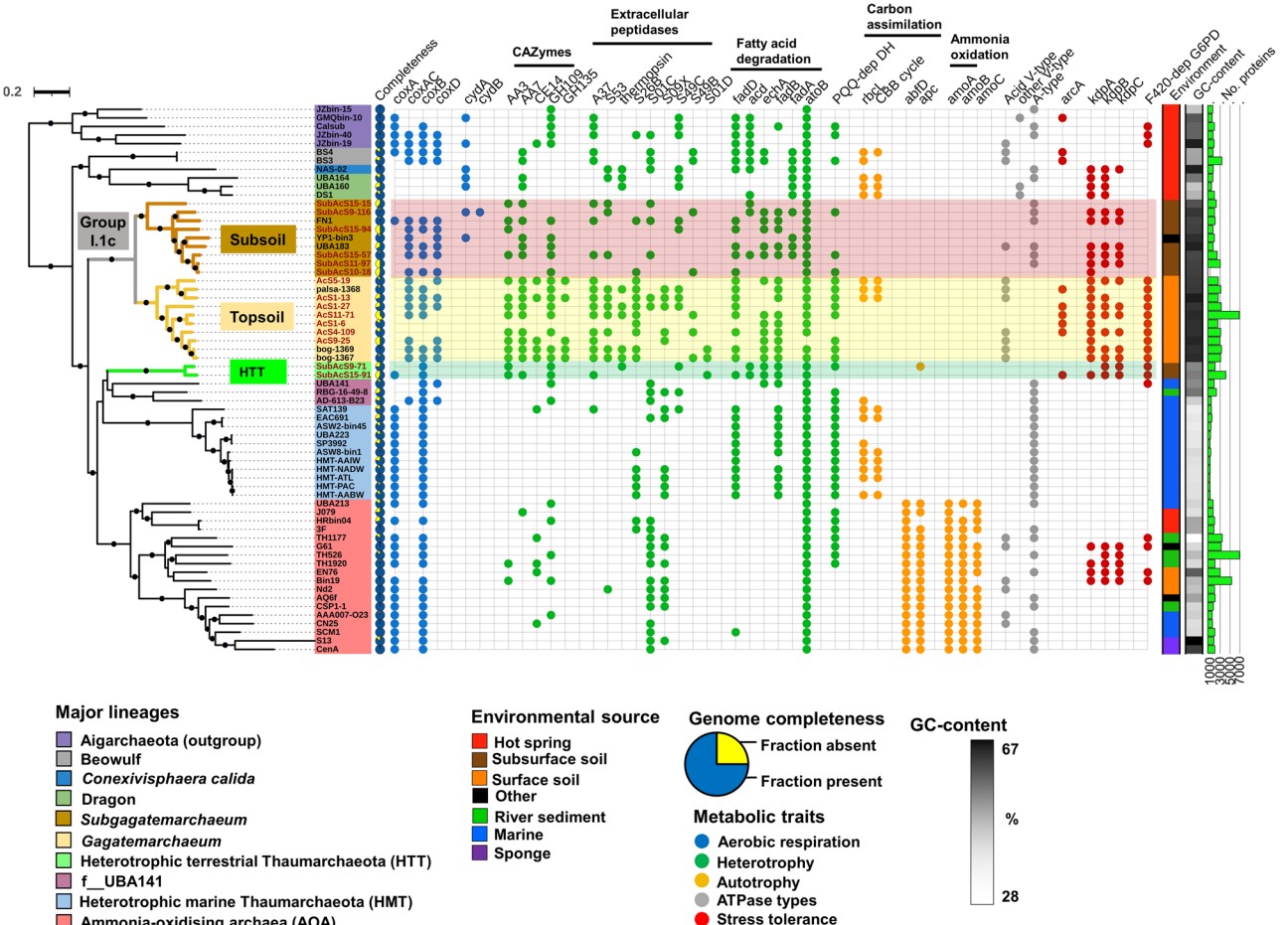

**Fig. 1 | Phylogenomic tree of Thaumarchaeota and distinctive traits of Gagatemarchaeaceae and Heterotrophic Terrestrial Thaumarchaeota genomes.** This tree comprises the major lineages of Thaumarchaeota, including 19 Gagatemarchaeaceae (previously Group I.1c Thaumarchaeota) genomes and two Heterotrophic Terrestrial Thaumarchaeota (HTT) genomes (Dataset 2). It was inferred by maximum likelihood reconstruction from 76 concatenated single-copy marker genes using the LG + C60 + G + F model. The 15 new genomes obtained in this study are labelled with the prefix "AcS" or "SubAcS" for topsoil and subsoil acidic soils, respectively. Dots indicate branches with 90% UFBoot and SH-aLRT support. The selected genes implicated in key ecosystem functions are: *cox*A (haem-copper oxygen reductases, subunit A; K02274), *cox*AC (haem-copper oxygen reductases, fused coxA and coxC subunit; K15408), *cox*B (haem-copper oxygen reductases,

subunit B; K02275), *cta*A (haem a synthase; K02259), *cta*B (haem o synthase; K02257), *cyd*A (cytochrome bd ubiquinol oxidase, subunit A; K00425), *cyd*B (cytochrome bd ubiquinol oxidase, subunit B; K00426), *rbc*L (ribulose-bisphosphate carboxylase large chain; K01601), CBB (Calvin–Benson–Bassham) cycle, *abf*D (4-hydroxybutyryl-CoA dehydratase; K14534), apc (acetyl-CoA/propionyl-CoA carboxylase; K18603), *amo*ABC (ammonia monooxygenase subunits A, B and C; K10944, K10945 and K10946), ATPase types (V/A-type ATPase, subunit A; K02117), *arc*A (arginine deiminase; K01478), *kdp*ABC (K+ transporting ATPase subunits A, B and C; K01546, K01547 and K01548), F$_{420}$-dep G6PD (F$_{420}$-dependent glucose-6-phosphate dehydrogenase; K15510) and PQQ-dep DH (PQQ-dependent dehydrogenase; PF13360).

---

(thermopsin), which are active at low pH[28–30]. Gagatemarchaeaceae genomes additionally encode the *liv* branched-chain amino acid transport system and multiple peptide and oligopeptide ABC transporter systems (Supplementary Data 10). They also encode genes for the degradation of amino acids alanine (*ala*, alanine dehydrogenase), glutamate (*gltBD*, glutamate synthase), aspartate (*aspB*, aspartate aminotransferase), serine (*ilvA*, threonine dehydratase), glycine (glycine cleavage system) and histidine (*hutHUI*) to precursor metabolites, as well as key genes involved in the degradation of branched-chain amino acids (Supplementary Data 10).

Members of this family also possess several genes involved in the beta-oxidation of fatty acids (Fig. 1), with most of the genomes encoding the long-chain acyl-CoA synthetase (*fadD*), required for initiated degradation of long-chain saturated and unsaturated fatty acids.

In contrast to the previous investigation of this clade using Fn1 as a representative genome[15], the aerobic respiration terminal oxidase (Complex IV) was detected in most Gagatemarchaeaceae genomes (14 of 19 genomes) (Fig. 1), suggesting that aerobic metabolism is common

in this family. The complex IV consists of a fused *coxA* and *coxC* subunit gene (*coxAC*), *coxB* and *coxD* genes. The *coxAC* genes of Gagatemarchaeaceae are members of the D- and K-channel possessing A1 subfamily of haem-copper oxygen reductases[31]. In addition, the microaerobic respiration terminal oxidase, cytochrome bd ubiquinol oxidase gene *cydA* was present in the *Subgagatemarchaeum* genomes, UBA183 and Fn1 (Fig. 1), suggesting adaptation of these organisms to environments where molecular oxygen is scarce. The cytochrome bd ubiquinol oxidases identified in UBA183 and Fn1 are members of the less common quinol:O2 oxidoreductase families qOR2 and qOR3 respectively, based on the cydA subfamily database[32].

Most Gagatemarchaeaceae possess the *Kdp* potassium transporter (EC:3.6.3.12), which is involved in pH homoeostasis in acidophiles by generating reverse membrane potential[33,34]. Half of the *Gagatemarchaeum* genomes encode an *arcA* arginine deiminase, which is involved in acid tolerance in several bacteria[35–37]. Additionally, all *Gagatemarchaeum* encode up to 12 copies of coenzyme F$_{420}$-dependent glucose-6-phosphate dehydrogenase (EC:1.1.98.2), which catalyses the conversion

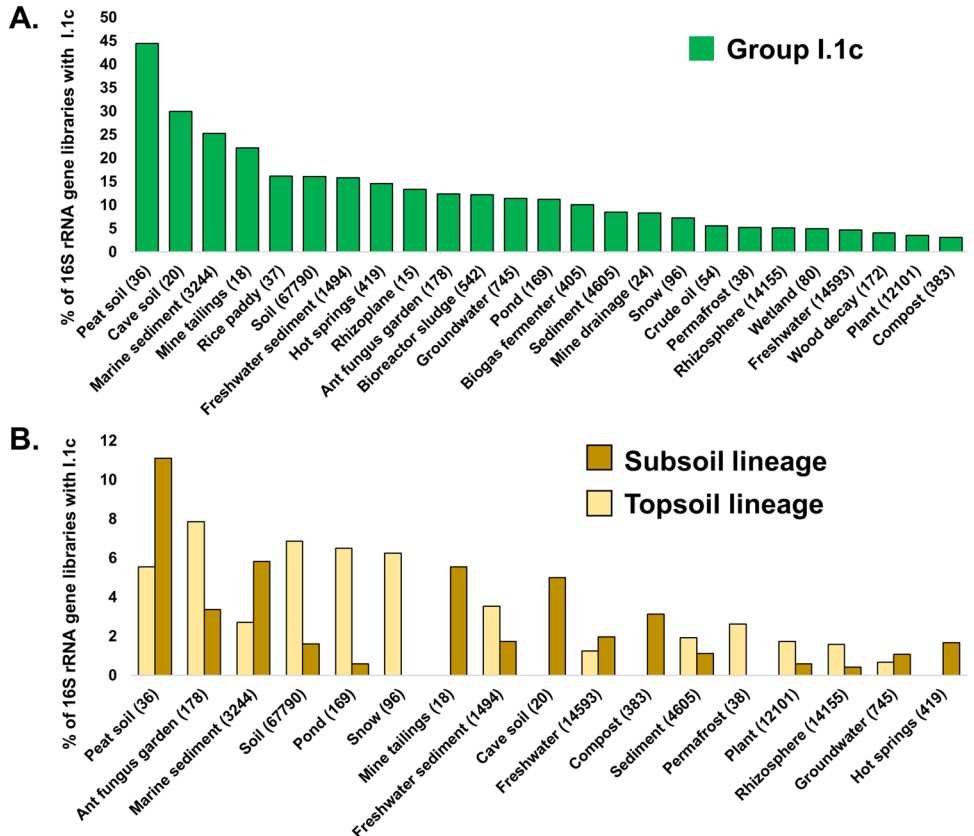

**Fig. 2 | Distribution of Gagatemarchaeaceae sequences in publicly available 16 S rRNA gene libraries, from a diverse range of environments. A** The assessment of family-level distribution was performed by querying the 16 S rRNA gene of bog-1369 against IMNGS[72] for reads of >400 bp that possessed >90% similarity. **B** The assessment of genus-level distribution was performed by querying the 16 S rRNA genes of bog-1369 (topsoil) and Fn1 (subsoil) against the collection of 16 S rRNA libraries in IMNGS[72] for reads >400 bp that possessed >95% sequence similarity. The number of samples of each environment is given in brackets. Source data are provided as a Source Data file.

of glucose-6-phosphate (G6P) to 6-phosphogluconolactone, with the subsequent reduction of the cofactor $F_{420}$ to $F_{420}H_2$, acting as a mechanism of resistance against oxidative stress[38] and nitrosative species[39,40]. Gene tree-species tree reconciliation and single-gene tree analysis of this gene family indicate that the multiple copies arose mainly from multiple progressive gene duplications throughout the evolutionary history of the *Gagatemarchaeum* (Supplementary Data 16 and 17, and Supplementary Fig. 3). Additionally, single-gene tree analysis indicates a second independent lateral acquisition of $F_{420}$-dependent glucose-6-phosphate dehydrogenase into the *Gagatemarchaeum* last common ancestor (LCA) (Supplementary Fig. 3). The high copy number of this gene family in *Gagatemarchaeum* suggests that these genes are metabolically important in topsoil colonisation.

Pyrroloquinoline quinone (PQQ)-dependent dehydrogenases catalyse the oxidation of a variety of alcohols and sugars[41]. These genes were highly expressed in marine environments and predicted to play an important physiological role in the heterotrophic marine Thaumarchaeota (HMT)[13,14]. PQQ-dependent dehydrogenases are also present in most *Gagatemarchaeum* (Fig. 1), with up to 8 genes per genome. This indicates that these genes may also play an important role in terrestrial non-AOA Thaumarchaeota. As noted for HMT[14], the PQQ-dependent dehydrogenases of *Gagatemarchaeum* tend to be colocalised on the genome, often appearing in adjacent pairs or trios (Supplementary Data 18). The PQQ-dependent dehydrogenases detected in this study formed 11 subfamilies (Supplementary Fig. 4). Four of the eight subfamilies detected in *Gagatemarchaeum* were also present in HMT[13]. Interestingly, PQQ-dependent dehydrogenases were also present in genomes of the Nitrososphaerales and Nitrosocaldales

lineages of AOA and could indicate an alternative energy source for these highly nutritionally specialised organisms. Gagatemarchaeaceae also lack marker genes of the archaeal, which is present in several AOA lineages[42], indicating that they are non-motile.

## Gagatemarchaeaceae genomic differences between topsoil and subsoil lineages

Despite the physiological similarities between members of this family, there were notable differences between the topsoil and subsoil lineages. There is strong evidence of lateral gene transfer in the energy-yielding V/A-type H+/Na+-transporting ATPases of these archaea. The topsoil lineages (*Gagatemarchaeum*) possess the acid-tolerant V-type ATPase and most subsoil lineages (*Subgagatemarchaeum*) encode the A-type ATPase (Fig. 1, Supplementary Fig. 5). The A-type ATPases have been previously predicted to be the ancestral thaumarchaeotal ATPase, with V-type ATPases being laterally acquired under environmental pressures such as low pH and high pressure[43]. Our analysis of the expanded Thaumarchaeota dataset supports this hypothesis, with multiple early diverging major lineages encoding the A-type ATPase (Supplementary Fig. 5).

*Gagatemarchaeum* genomes also possess significantly more CAZymes (involved in carbohydrate degradation) ($P < 0.02$) and peptidases (involved in peptide degradation) ($P < 0.02$) than *Subgagatemarchaeum* genomes (Fig. 3). In addition to functional gene differences, the topsoil and subsoil lineages vary in their genome characteristics. *Gagatemarchaeum* genomes (median 2.5 Mb; range 2.2–3.4 Mb) are, on average, 47% larger than *Subgagatemarchaeum* members (median 1.7 Mb; range 1.2-2.5 Mb) ($P < 0.001$) and have slightly

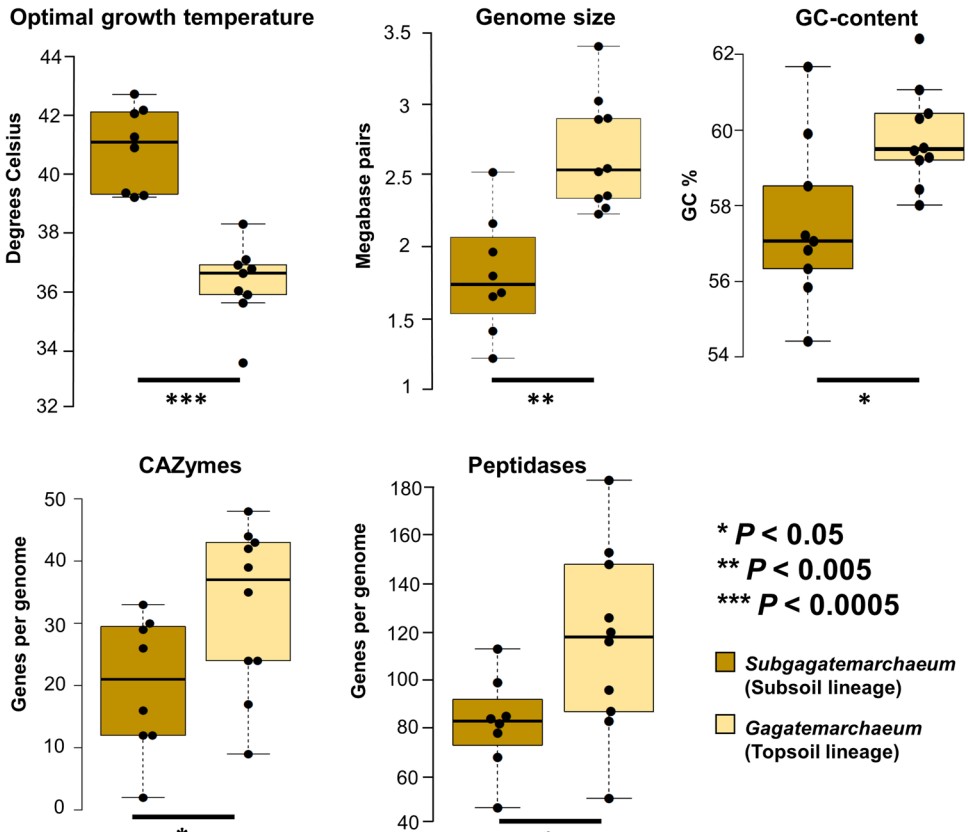

**Fig. 3 | Genomic differences between topsoil and subsoil Gagatemarchaeaceae lineages.** Optimal growth temperature (estimated theoretically on the entire proteome) ($P = 3.23e^{-5}$), genome size ($P = 5.15e^{-4}$), GC-content ($P = 0.02$) and CAZyme ($P = 0.04$) and peptidase ($P = 0.03$) numbers were compared for the two lineages, with adjustment for genome completeness when required. Dots indicate data points ($n = 19$ genomes), centre lines show the medians, box limits indicate the 25th and 75th percentiles and whiskers extend 1.5 times the interquartile range from the 25th and 75th percentiles. Significant differences between the two groups were estimated with two-tailed unequal variance (Welch's) t-tests, with the significance level indicated below each graph. Source data are provided as a Source Data file.

higher GC-content ($P < 0.02$) (Fig. 3). Additionally, the predicted optimal growth temperature of the *Subgagatemarchaeum* (average 40.5 °C) was slightly higher than that of *Gagatemarchaeum* (average 36 °C) ($P < 1e^{-5}$) (Fig. 3). Values for CAZymes, peptidases and genome size in this comparison have been adjusted by the genome incompleteness.

## Metabolism of the heterotrophic terrestrial Thaumarchaeota (HTT) clade

Two newly acquired genomes, representing a novel terrestrial genus related to the HMT and the uncharacterised f_UBA141 family (Fig. 1), lack the gene markers for autotrophic carbon fixation (Supplementary Data 10) and possess genes for carbohydrate, peptide, and fatty acid utilisation (Fig. 1). The *coxA* gene in SubAcS9-71 is a member of the B subfamily of haem-copper oxygen reductases, in contrast to the A2 subfamily genes found in HMT and the A1 subfamily genes found in AOA and Gagatemarchaeaceae. This indicates that the Complex IV of f_UBA184 and HMT families were independently acquired. HTT also possess acid tolerance genes such as the *Kdp* potassium transporter (EC:3.6.3.12) present in Gagatemarchaeaceae and terrestrial AOA[1] or the *arc*A arginine deiminase present in *Gagatemarchaeum*. Two PQQ-dependent dehydrogenases of the 4.1 subfamilies (Supplementary Fig. 4) are present in the HTT genome SubAcS15-91, indicating another heterotrophic energy source for these terrestrial organisms.

## Genome evolution of the non-ammonia oxidising Thaumarchaeota

The 15 newly acquired genomes and the recent description of other non-AOA Thaumarchaeota[11,13,14] allow us to address some of the open questions about genome evolution in Thaumarchaeota, including the temperature preference of the AOA ancestor (Fig. 4). Ridge regression of extant genome optimal growth temperatures (OGTs) across the thaumarchaeotal species tree indicates that the thaumarchaeotal LCA had an OGT of 48 °C, with a gradual reduction in OGT to 43 °C for the AOA LCA (Fig. 4). Our analysis predicts that the AOA and multiple lineages of non-AOA Thaumarchaeota form a mesophilic clade, except for some thermophilic genomes belonging to the Nitrosocaldales lineage. The non-AOA Thaumarchaeota lineage encompassing the Dragon (DS1, UBA164 and UBA160), Beowulf (BS3 and BS4) and *Conexivisphaera calida* NAS-02 genomes is sister to this mesophilic clade. This reconstruction supports the hypothesis that the LCA of AOA was a mesophile[1], which was hypothesised based on the presence of mesophilic Nitrosocaldales genomes (Thaumarchaeota archaea SAT137 and UBA213) and related non-AOA Thaumarchaeota lineages. The current increased representation of mesophilic non-AOA Thaumarchaeota lineages provides a scenario which contradicts the earlier hypothesis of thermophilic archaeal ammonia oxidation ancestor[42,44,45]. A previous study predicted the reverse gyrase, *rgy*, (considered a hallmark enzyme of thermophily in prokaryotes[46,47]) to be present in the AOA LCA[42]. However, gene tree analysis indicates that the *rgy* gene present in the Nitrosocaldales genome J079 (Supplementary Data 10), was acquired recently (Supplementary Fig. 6), consistent with the theory that the AOA LCA was a mesophile.

The genome GC content varies significantly across the Thaumarchaeota phylum (range 29–67%). The genomes of the HMT clade have a low GC content (range 31–34%), consistent with most lineages of AOA (Fig. 1). GC content is higher in the genomes of the HMT-related

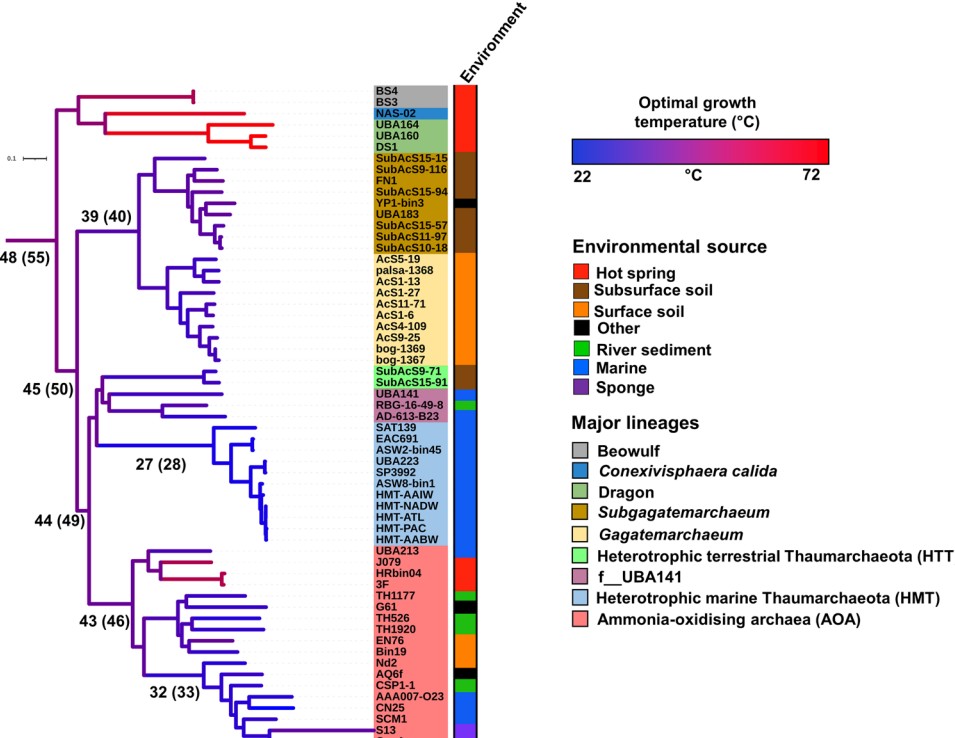

**Fig. 4 | Sequence-based prediction of thermal adaptation throughout Thaumarchaeota history.** The optimal growth temperature (OGT) of extant organisms was predicted based on genome sequences using Tome[78], with previously demonstrated good correspondence between predicted and empirical values[1]. Ancestral OGTs were predicted using a ridge regression approach[79] across the Thaumarchaeota phylum tree (Dataset 3) using the extant organism OGTs as leaf values. Branches were coloured based on their predicted OGT, and key ancestral OGTs were presented on specific branches. Values presented in parenthesis are included to give overestimated values of OGT (as described in Methods) without altering the conclusion.

clades (GC 46–49%) and even higher in the Gagatemarchaeaceae (range 54–62%). This is consistent with previous observations of higher GC content in terrestrial than related aquatic species[48,49].

### Evolution of Thaumarchaeota Group I.1c topsoil and subsoil lineages

The Gagatemarchaeaceae have larger genomes than other non-AOA Thaumarchaeota lineages, especially the topsoil lineage - the *Gagatemarchaeum* (Fig. 1). A gene tree-species tree reconciliation approach was adopted to study the mechanisms influencing the evolution of the Gagatemarchaeaceae family and decipher if the large genome size results from a reduction of the ancestral genome in other lineages or genome expansion in Gagatemarchaeaceae (Supplementary Data 19 and Supplementary Fig. 7). As observed in Nitrososphaerales[1] and Cyanobacteria[50], genome expansion occurred during the transition into terrestrial environments (Supplementary Fig. 8). Genome expansion was likely initiated by numerous intra- and inter-phyla gene transfers, with the latter being crucial for providing novel metabolic acquisition, enabling environmental transition. Two periods of extensive acquisition of novel gene families (by inter- and intra-phylum gene transfer) were predicted in the early evolution of Gagatemarchaeaceae (Fig. 5A). The first was in the Gagatemarchaeaceae LCA (282 inter- and 342 intra-phylum gene transfers), and the second was in the *Gagatemarchaeum* LCA (214 inter and 151 intra-phylum gene transfers). ALE[51], the reconciliation tool used for inferring these transfers employs a probabilistic model of gene duplication, transfer and loss, averaging over the uncertainty in the gene tree and the uncertainty in the mapping of gene tree branches to the species tree (the reconciliation). Inferred numbers of events therefore represent averages of over 100 sampled reconciliations for each gene family[52]. While the method accounts for phylogenetic uncertainty, it does make use of topological information so stochastic gene tree error or artifacts such as long

branch attraction have the potential to moderately inflate the number of inferred transfers[52]. High levels of gene duplication were also detected throughout the evolution of the *Gagatemarchaeum* genus (Fig. 5B), further driving genome expansion. These duplications include the newly acquired gene families, of which 10–20% are present in multiple copies in extant *Gagatemarchaeum* genomes (Supplementary Data 20).

Gene losses in the Gagatemarchaeaceae lineages were higher than in the rest of the phylum ($P < 0.04$) (Supplementary Fig. 9), but generally, losses were less punctuated (i.e., events were less concentrated in a small number of species tree branches, as indicated by a lower punctuation score[1,2]) across the phylum history than the other mechanisms of gene content change (Supplementary Fig. 10). The Gagatemarchaeaceae LCA received a notable influx of genes through lateral transfer from other members of the Thaumarchaeota (342 genes) (Supplementary Fig. 9). Over a third (38%) of these incoming genes were predicted to have been transferred from the lineages f_UBA-141 and HTT (Supplementary Data 21).

The Gagatemarchaeaceae LCA gained many key genes relevant for their adaptation to soil environments, including seven peptidases (families S33, S09X, M95, N11, S33, M50B and M03C) and four genes involved in the utilisation of myo-inositol (*iolB, C, E* and *G*), an abundant chemical in soil that can be used as a sole carbon source by diverse bacteria[53] (Fig. 6). This LCA also gained three genes involved in inosine monophosphate biosynthesis (*purD, H* and *M*), which metabolically link the pentose phosphate pathway and histidine metabolism to the production of purines. The *Kdp* potassium transporter (EC:3.6.3.12), likely implicated in acidophily, was also acquired by this LCA. Other gene gains in this LCA included the *Pnt* NAD(P) transhydrogenase (EC:1.6.1.2), which performs the reversible transfer of electrons from NADH to NADP[54], and $F_{420}H_2$:NADPH oxidoreductase (EC:1.5.1.40), which transfers electrons from NADPH to oxidised

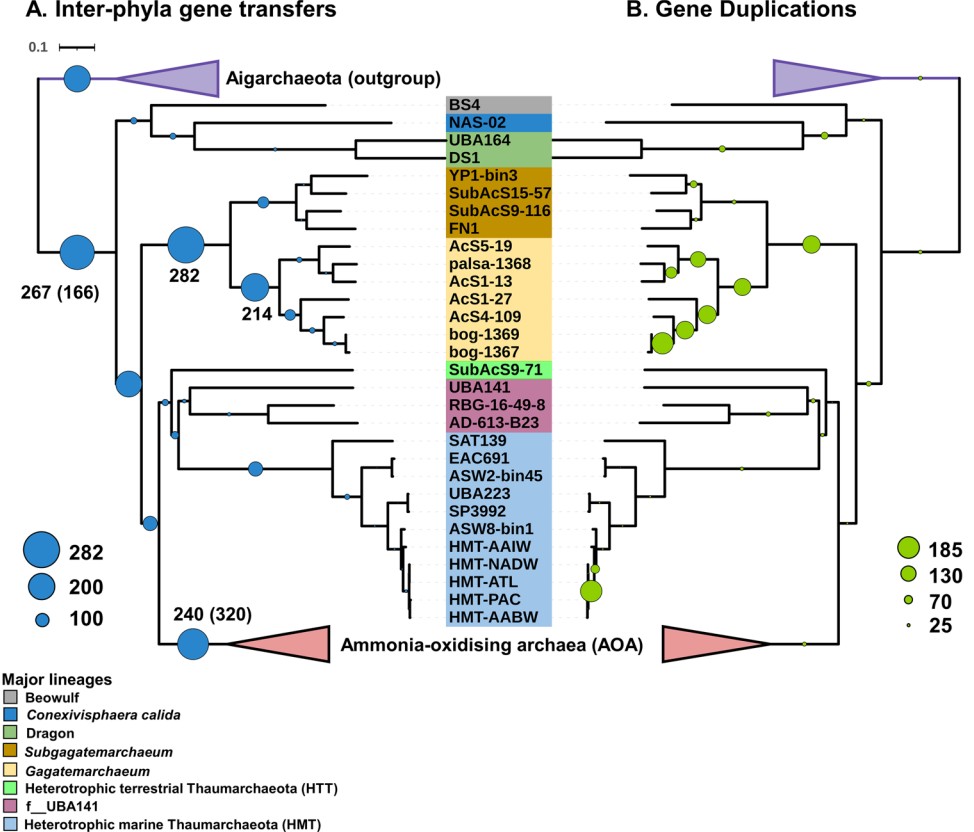

**Fig. 5 | Acquisition of novel gene families (through inter-phyla gene transfers) and gene duplications along non-AOA Thaumarchaeota lineages.** The quantitative and qualitative predictions of inter-phyla gene transfers (**A**) and gene duplications (**B**) were estimated across the Thaumarchaeota phylum tree (Dataset 3) using a gene tree-species tree reconciliation approach as described in the Methods section "Predicting gene content changes across evolutionary history". Scale numbers indicate the range of the predicted number of events for a given mechanism, and circle sizes are proportional to the number of events.

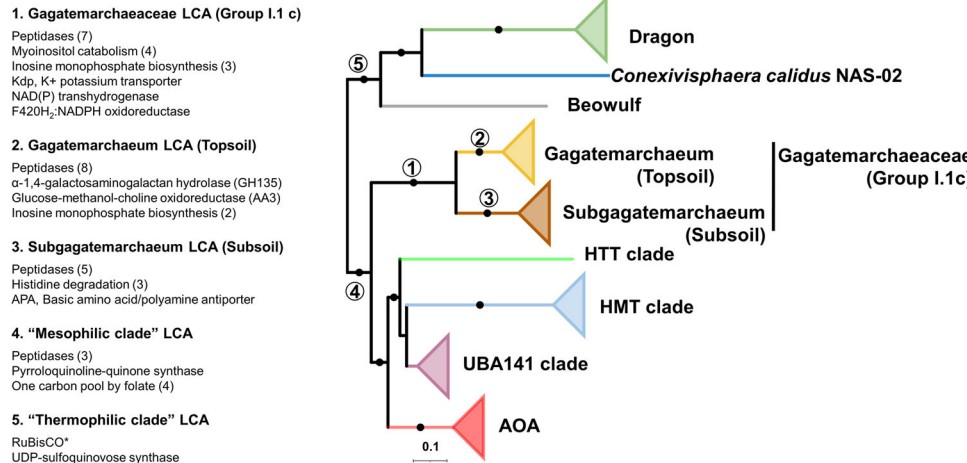

**Fig. 6 | Notable functional gene gains during non-AOA Thaumarchaeotal evolution.** The gain of gene families in specific ancestors was predicted by comparing the probabilistic ancestral gene content reconstructions. Dots indicate branches with at least 95% UFBOOT and SH-aLRT support. Circled numbers indicate specific thaumarchaeotal ancestors. The numbers in parentheses correspond to the number of gene families of the named function gained in each ancestor. *The *rbc*L gene was slightly below the 0.5 reconciliation copies threshold (0.43), but we decided to retain it in the analysis.

coenzyme F$_{420}$[55] (Fig. 6), indicating an important role for electron transfer between redox cofactors in these organisms.

The *Gagatemarchaeum* LCA gained eight peptidases (three S33, S01D, C44, two S09X and S49C), two additional genes involved in inosine monophosphate biosynthesis (*purB* and *E*) and the α-1,4-galactosaminogalactan hydrolase (GH135), which is potentially involved in fungal cell wall degradation (Fig. 6). This LCA also gained a member of the Glucose-methanol-choline oxidoreductase family (AA3), while several family members are present in diverse Gagatemarchaeaceae. These enzymes catalyse the oxidation of alcohols or carbohydrates in lignocellulose degradation of wood-degrading fungi[56], but their function in archaea has not been studied.

The *Subgagatemarchaeum* LCA gained the histidine degrading genes *hutH*, *U* and *I*, with the coinciding loss of the histidine

biosynthetic genes *hisD*, *F*, *G* and *H*. Indeed, analysis of extant genomes indicates that this lineage is incapable of biosynthesising histidine (Fig. 6). This suggests that *Subgagatemarchaeum* uptakes extracellular histidine, which is at least partially used as a source of energy, carbon, and nitrogen. Like the family's LCA, the *Subgagatemarchaeum* LCA also gained multiple peptidases (two M38, M32, C26 and S33 family peptidases).

The LCA of the "mesophilic clade" of Thaumarchaeota, which excludes the thermophilic *Conexivisphaera*, Dragon and Beowulf clades, was predicted to have gained three peptidases (M61, M48B and C26) (Fig. 6). It also gained the PQQ synthase, *pqq*C, which performs the final steps in PQQ biosynthesis[57], reflecting the abundance of PQQ-dehydrogenases in its descendants (Fig. 1). This LCA also gained four genes of the one-carbon metabolic pathway. Notably, both the 5,10-methylene-tetrahydrofolate dehydrogenase/cyclohydrolase (FolD) and 10-formyltetrahydrofolate synthetase (Fhs) pathways for N10-formyltetrahydrofolate (metabolite in initiator tRNA and purine nucleotide biosynthesis) production were gained (Fig. 6). In *Escherichia coli*, *fhs* provides a selective advantage under anaerobic conditions, particularly in the presence of formate[58]. Therefore, the possession of both pathways may indicate adaptation to a facultatively anaerobic strategy in the mesophilic clade.

Although the evolution of the more "thermophilic clade" of non-AOA Thaumarchaeota (*Conexivisphaera*, Dragon and Beowulf) was not studied here in detail due to a lack of representative genomes, its LCA acquired the ribulose-bisphosphate carboxylase large chain, *rbcL*, which was shown previously to classify as a Type III RuBisCO[11] (involved in carbon fixation) (Fig. 6). It also acquired a UDP-sulfoquinovose synthase, an essential gene in the production of sulfolipids[59], which reduces the microbial phosphate requirements in oligotrophic marine environments[60].

### Notes
Gagatemarchaeaceae, *Gagatemarchaeum* and *Subgagatemarchaeum* are not validly published names under the International Code of Nomenclature of Prokaryotes and thus can be considered as candidatus taxa. The *Candidatus* prefix was omitted from these taxa in the manuscript for brevity.

## Discussion
A previous genomic analysis of a single Gagatemarchaeaceae genome, Fn1, indicated that this group of organisms is anaerobic[15]. However, subsequent experimental evidence suggested that Gagatemarchaeaceae grow under aerobic conditions in soil[16]. Our genomic analysis detected the presence of microaerophilic respiration genes in Fn1 and revealed the presence of genes for aerobic respiration in most Gagatemarchaeaceae genomes, corroborating the empirical aerobic growth.

The Gagatemarchaeaceae appears to have undergone an early bifurcation in its evolutionary history, with the *Gagatemarchaeum* and *Subgagatemarchaeum* genera adapting to topsoil and subsoil soils, respectively. This divergence corresponds with some notable metabolic and genomic differences between the two lineages. *Gagatemarchaeum* genomes have significantly more genes for utilising exogenous organic substrates, such as carbohydrates and proteins, than their subsoil sister lineage genomes. They also possess genes for acid tolerance that were absent from the *Subgagatemarchaeum* genomes. *Gagatemarchaeum* genome sizes and GC-contents are greater than that of *Subgagatemarchaeum*, likely due to better adaptation to fluctuating environments[61] and involvement with resistance to DNA damage[49], respectively. Together, this suggests an adaptation of *Gagatemarchaeum* to a nutrient-rich but environmentally stressed lifestyle in topsoils, contrasting the *Subgagatemarchaeum* nutrient-poor lower-stress lifestyle in subsoils.

The larger genomes observed in Gagatemarchaeaceae were driven by early lateral gene acquisition and subsequent gene duplication in topsoil lineages. This paradigm of genome expansion has been observed previously in the terrestrial AOA Nitrososphaerales[1], indicating that gene duplication may be a common mode of genome expansion in archaea. This paradigm of early gene acquisition followed by extensive duplication has been proposed as the mechanism through which early eukaryotes increased in complexity, thereby differentiating from their archaeal ancestor[62,63]. Our work suggests that this paradigm has broader implications than in the archaeal-eukaryote branch of life.

Previous gene tree-species tree reconciliation studies have examined genome evolution during expansion into drastically different ecosystems, such as from aquatic to terrestrial[1] or to hypersaline environments[3], but this is (to the best of our knowledge) the first study to use these techniques to study transitions into more similar and spatially related ecosystems, representing a majority of the habitat expansions. The phylogenetic approach cannot distinguish if the Gagatemarchaeaceae LCA inhabited a topsoil environment and an early diverging member expanded into subsoil soils or vice versa, but the higher rate of gene family loss between Gagatemarchaeaceae LCA and subsoil genomes might suggest the former, to the extent that the loss of ancestral genes might be associated with habitat shift.

Contrasting theories have been proposed about the thermal preference of the AOA LCA, suggesting either a hyperthermophilic[42,44,45] or a mesophilic ancestor[1]. When initially proposed[44], the hyperthermophilic ancestor hypothesis was in good agreement with available data, including an early branching hyperthermophilic AOA (*Nitrosocaldus yellowstonii*) and hyperthermophilic closest relatives to the Thaumarchaeota. Since then, multiple major lineages of non-AOA Thaumarchaeota mesophiles have been discovered in this work and previously, including Gagatemarchaeaceae[15], HMT[13,14], HTT and mesophilic Nitrosocaldales[1]. These mesophilic lineages were not included in this early work[44] or some subsequent predictions of AOA LCA thermal preference[42,45]. Based on this expanded sampling of taxonomically diverse thaumarchaeotal genomes, our predictions suggest that the transition to a mesophilic state occurred earlier in Thaumarchaeota evolution than in the AOA LCA.

## Methods
### Sampling, sequencing and metagenomic assembled genome creation
Soil samples were collected from nine sites in Scotland (UK) (Supplementary Data 6), and the environmental DNA was extracted using Griffith's protocol[64] with modifications[65]. DNA libraries were prepared using Illumina TruSeq DNA PCR-Free Library Prep Kit with one μg of environmental DNA. The sequencing was performed on the Illumina NovaSeq S2 platform ($9.2 \times 10^{10}$ bases per sample on average, Macrogen company, Supplementary Data 6), generating 150 bp paired-end reads. Reads were filtered using the READ_QC module[66], and high-quality reads for each metagenome were assembled using MEGAHIT[67]. Binning of resulting contigs was performed with MaxBin2[68] and metaBAT2[69], and the results were consolidated using the Bin_refinement module from MetaWRAP[66]. Completeness and contamination of bins were estimated with CheckM 1.0.12[70], and bins with completeness >45% and contamination <10% were retained for further analysis. Genome coverage by metagenomic reads was calculated using CoverM v0.6.1 (https://github.com/wwood/CoverM). The relative abundance of each genome was estimated by competitive read recruitment of metagenomics reads to genome sequences using the Quant_bins module in metaWRAP[66]. Differences in genome abundance between topsoil and subsoil metagenomes were validated by either one- or two-tailed unequal variance (Welch's) *t*-tests. Taxonomic characterisation to genus level was performed using the classify_wf function in GTDB-Tk v1.7.0[71] using the R202 GTDB release. Average amino acid identities (AAIs) between pairs of genomes were calculated using CompareM (https://github.com/dparks1134/CompareM), and species were defined with AAI thresholds higher than 95%.

### Collection of public genomes

Forty-four publicly available thaumarchaeotal genomes were selected from previous literature. This included 19 non-AOA Thaumarchaeota previously used in a detailed thaumarchaeotal phylogenomic analysis[1], seven heterotrophic deeply-rooted marine Thaumarchaeota genomes[13,14], the *Conexivisphaera calida* genome[12], four genomes classified in GTDB as members of the families f_UBA183 (Gagatemarchaeaceae; Group I.1c) and f_UBA141, and a selection of 18 genomes representing the major lineages of ammonia-oxidising archaea (AOA)[1]. These genomes were downloaded from NCBI (www.ncbi.nlm.nih.gov).

### Prevalence of Gagatemarchaeaceae in public 16 S rRNA libraries

The bog-1369 and Fn1 genomes were chosen as representative organisms as these genomes meet the quality criteria for type material suggested for MIMAGs[22,23], as detailed later in the manuscript. The 16 S rRNA gene of the genome bog-1369 was used to represent the Gagatemarchaeaceae family and queried against the extensive collection of 16 S rRNA gene libraries in IMNGS[72] for reads ≥400 bp presenting ≥90% sequence similarity. Additionally, the 16 S rRNA genes of the bog-1369 and Fn1 genomes were used to represent the topsoil and subsoil lineages of Gagatemarchaeaceae, respectively. They were queried against IMNGS for reads ≥400 bp that possessed ≥ 95% sequence similarity. Prevalence was calculated as the percentage of samples of a given environment where Gagatemarchaeaceae was detected.

### Group I.1 c Thaumarchaeota classification

The 16 S rRNA gene sequences were extracted from Gagatemarchaeaceae genome sequences using Barrnap v0.9 (--kingdom arc, archaeal rRNA) (https://github.com/tseemann/barrnap) and combined with previously published 16 S rRNA sequences used for a Group I.1c phylogenetic tree[19] (https://github.com/SheridanPO-Lab/I.1c-Group). A phylogenetic tree was constructed with IQ-TREE 2.0.3[73] using the SYM + R5 model. The Gagatemarchaeaceae genomes for which 16 S rRNA genes could be recovered were directly compared to the previously published taxonomic classification. The classification of several Gagatemarchaeaceae genomes without 16 S rRNA gene sequences was inferred based on their phylogenomic relationships to genomes with 16 S rRNA gene sequences.

### Determination of genome characteristics

All genomes were annotated using Prokka v1.14[74], and GC content and genomic size were calculated using QUAST[75]. Environmental source information and genome sequence type (i.e. culture, SAG, MAG, etc.) were retrieved from NCBI or associated published studies. Protein novelty[1,2], defined as the percentage of encoded proteins that lack a close homologue (e-value < 10 − 5, % ID > 35, alignment length > 80 and bit score > 100) in the arCOG database[76], was estimated using Diamond BLASTp[77]. Optimal growth temperatures (OGT) were predicted in silico for each genome (based on Tome[78], which uses a machine-learning model of amino acid dimer abundance in all genes of a genome). OGT in ancestors of extant Thaumarchaeota was inferred with RidgeRace[79], which uses ridge regression for continuous ancestral character estimation and uses the Tome predictions as leaf values. RidgeRace was previously used for predicting pH preference in thaumarchaeotal ancestors[80]. As Tome has been shown to underestimate the OGT of hyperthermophilic organisms[1], OGT values for key ancestors were also estimated using a 19 °C-increased OGT for all genomes presenting a predicted OGT greater than 45 °C. The 19 °C value was selected as the highest observed discrepancy between Tome predictions and experimental predictions[1].

### Gene marker selection and phylogenomic inference

For each dataset, ortholog groups (OGs) were detected using Roary (-i 50, -iv 1.5)[81]. Core OGs were defined as those present in a single copy in

each genome and present in at least 70% of the genomes. Core OGs were aligned individually using MAFFT L-INS-i[82], and spurious sequences and poorly aligned regions were removed with trimAl (automated1, resoverlap 0.55 and seqoverlap 60)[83]. Alignments were removed from further analysis if they presented evidence of recombination using the PHItest[84]. The remaining alignments were concatenated into a supermatrix for each dataset. Maximum-likelihood trees were constructed for each dataset supermatrix with IQ-TREE 2.0.3[73] using the complex mixture model LG + C60 + G + F. Branch supports were computed using the SH-aLRT test[85] and 2000 UFBoot replicates. A hill-climbing nearest-neighbour interchange (NNI) search was performed to reduce the risk of overestimating branch supports.

### Phylogenomic analysis

This study used three separate genome datasets for different analyses (Supplementary Data 22). Dataset 1 consisted of 19 Gagatemarchaeaceae genomes, two UBA141-like genomes and three AOA. Dataset 2 consisted of 64 genomes of Thaumarchaeota and related species (completeness > 45%, contamination < 10%). Dataset 3 consisted of 52 higher-quality genomes of Thaumarchaeota and closely related species (completeness > 70%, contamination <5%). Dataset 2 was used to infer the phylogenomic tree presented in Figs. 1 and 4 (note: The same Group I.1c phylogenetic topology was obtained using Dataset 1). Dataset 3 was used to infer the phylogenomic tree presented in Figs. 5 and 6.

### Predicting gene content changes across evolutionary history

For the higher-quality genomes dataset (Dataset 3; 52 genomes, completeness > 70%, contamination < 5%), gene families were inferred with Roary 3.12.0[81] with low stringency (-i 35, −iv 1.3, −s). Sequences shorter than 30 amino acids and families with less than four sequences were removed from further analysis. All remaining sequences within each family were aligned using MAFFT L-INS-i 7.407[82], and poorly aligned sites were removed with trimAl 1.4.1 ("automated1" setting)[83]. Individual ML phylogenetic trees were constructed for each alignment with IQ-TREE 2.0.3[73] using the best-fitting protein model predicted in ModelFinder[86].

Each gene family tree was probabilistically reconciled against the previously created rooted supermatrix tree (Dataset 3) using the ALEml_undated algorithm of the ALE package[51]. For the gene family trees being probabilistically reconciled against the species tree (3914 of 3921), this approach allowed inferring the numbers of duplications, intra-LGTs, losses and originations (inter-LGTs) on each branch of the species tree. A 0.5 reconciliation copies threshold[1] was used to determine a gene family's presence in ancestral gene content reconstructions. Genome incompleteness was probabilistically accounted for within ALE using the genome completeness values estimated by CheckM 1.0.12[70]. The mechanism of gene content change on every branch of the species tree was estimated using branchwise_numbers_of_events.py, as described before[1]. The number of intra-LGTs transferring into and transferring from every branch of the species tree was estimated with calc_from_to_T.sh, as described before[2]. All phylogenomic trees were visualised using iTOL[87].

### Functional annotation of genomes

Genomes were annotated with the KEGG database[88] using GhostKOALA[20], with the arCOG database[76] using Diamond BLASTp[77] (best-hit and removing matches with e-value > 10 − 5, % ID < 35, alignment length <80 or bit score <100) and with the Pfam[89] database using hmmsearch[21] (HMMER v3.2.1) (-T 80). The subfamily classification of *cyd*A was performed using hmmsearch (-T 80) with the *cyd*A subfamily database[32]. The subfamily classification of *cox*A genes was performed using the haem-copper oxygen reductase database[90]. Carbohydrate-active enzymes were annotated using profile HMM from dbCAN (http://bcb.unl.edu/dbCAN2/) (filtered with hmmscan-

parser.sh and by removing matches with mean posterior probability < 0.7). Peptidases were annotated using Pfam profile HMMs corresponding to MEROPs families, as described previously[91]. Extracellular carbohydrate-active enzyme peptidases were identified using Signalp 5.0[92] (-org arch, archaeal signal peptides) to detect the presence of signal peptides. The presence of motility genes in Gagatemarchaeaceae was initially assessed by the presence of the conserved archaellum subunits C (arCOG05119), D/E (arCOG02964), F (arCOG01824), G (arCOG01822) and J (arCOG01809). The 5 S, 16 S and 23 S rRNA and tRNA genes were identified using Barrnap v0.9 (--kingdom arc, archaeal rRNA) (https://github.com/tseemann/barrnap) and tRNAscan-SE v2.0.5[93] (-A, archaeal tRNA), respectively. The 16 S rRNA genes from the different genomes were compared by a pairwise analysis using BLASTn v2.9.0[18].

### Single gene tree analysis

To infer a phylogeny for F420-dependent glucose-6-phosphate dehydrogenase genes, an expanded inter-domain set of prokaryotic genomes (Supplementary Data 17) was annotated against the KEGG database[88] using GhostKOALA[20], and the protein sequences of all genes annotated as F420-dependent glucose-6-phosphate dehydrogenase were extracted and combined with the F420-dependent glucose-6-phosphate dehydrogenase genes detected in this study. To infer a phylogeny for the Thaumarchaeota PQQ-dependent dehydrogenases, protein sequences were extracted for genes annotated as PQQ-dependent dehydrogenases by their possession of the PF13360 conserved domain. To infer a phylogeny for the V/A-ATPase genes detected in this study, protein sequences of the three largest subunits of V/A-ATPase (atpA, atpB and atpI) extracted from the genomes in Dataset 1 and combined with those analysed in a previous study of Lutacidiplasmatales ATPases[2]. All subunits were individually aligned and then concatenated into a single partitioned supermatrix. This aligned supermatrix is available at https://github.com/SheridanPO-Lab/I.1c-Group/tree/main/Alignments with the filename "ATPase_supermatrix.aln".To infer a phylogeny for the reverse gyrase genes detected in the study, protein sequences that possessed the IPR005736 domain were downloaded from UniProt[94]. These sequences were clustered with CD-HIT[95] using an identity threshold of 50%. Representative protein sequences from each cluster and thaumarchaeotal reverse gyrases were combined into a single dataset. Each of these multi-protein sequence datasets were aligned using MAFFT L-INS-i[82], and spurious sequences and poorly aligned regions were removed with trimAl (automated1)[83]. Maximum-likelihood trees were constructed for each alignment with IQ-TREE 2.0.3[73] using the best-fitting model in ModelFinder[86]. Branch supports were computed using 1000 UFBoot replicates. A hill-climbing nearest-neighbour interchange (NNI) search was performed to reduce the risk of overestimating branch supports. The resulting trees were rooted using minimal ancestor deviation[96]. Subfamilies of the Thaumarchaeota PQQ-dependent dehydrogenases were determined by the average pairwise distance between leaves using TreeCluster[97].

### Reporting summary

Further information on research design is available in the Nature Portfolio Reporting Summary linked to this article.

## Data availability

Accession numbers for the 15 new genomes presented in this study can be found in Supplementary Data 1 and under the NCBI BioProject PRJNA883052. The accession numbers for publicly available genome sequences used in the phylogenomic genome datasets can be found in Supplementary Data 22 and accessions for the expanded inter-domain set of prokaryotic genomes, used for single gene tree analysis, can be found in Supplementary Data 17. Public data is available from NCBI,

KEGG, dbCAN, arCOG, PFAM, TIGRFAM and GTDB R202. Source data are provided in this paper.

## Code availability

Scripts for general manipulation of ALE outputs have been deposited at https://github.com/Tancata/phylo/tree/master/ALE (https://doi.org/10.5281/zenodo.4012549)[98], and additional scripts, alignments and phylogenies specific to this work have been deposited at https://github.com/SheridanPO-Lab/I.1c-Group (https://doi.org/10.5281/zenodo.8421019)[99] and https://github.com/SheridanPO-Lab/ALE_analysis (https://doi.org/10.5281/zenodo.8421034)[100].

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

## Acknowledgements

UKRI financially supported P.O.S and Y.M. through the NERC grant (NE/R001529/1). In addition, C.G.-R. and T.A.W. were supported by Royal Society University Research Fellowships (URF150571 and UF140626, respectively). We thank Tony Travis for his support with Biolinux. The authors would also like to acknowledge the support of the Maxwell computer cluster funded by the University of Aberdeen.

## Author contributions

P.O.S., T.A.W. and C.G.-R. designed the study and developed the theory. P.O.S. collected the samples and Y.M. performed DNA extraction. P.O.S. assembled the 15 new genomes and performed genomic analyses. P.O.S., T.A.W. and C.G.-R. interpreted the results and wrote the paper. All authors have accepted the final version of the paper.

## Competing interests

The authors declare no competing interests.
