## [Peer Review File · Nature Communications]

Genomics of soil depth niche partitioning in the Thaumarchaeota family GagatemarchaeaceaeReviewers' Comments:

Reviewer #1:

Remarks to the Author:

The study of Sheridan et al. focuses on the evolution of deeply rooted non-ammonia oxidizing Thaumarchaeota (currently reclassified as class Nitrososphaeria) and specifically the mechanisms of genomic evolution underpinning the diversification of lineages that colonized terrestrial subsoil and topsoil niches. The authors assemble 15 novel metagenomes from topsoil and subsoil environments, and through phylogenomic analyses propose that the lineage of I.1c Thaumarchaeota be classified as a novel archaeal family, named *Ca. Gagatemarkarchaeaceae*, comprising two genera (*Ca. Gagatemarkarchaeum* and *Ca. Subgagatemarkarchaeum*) and 17 species. Through comparative genomics, they characterise the putative metabolism and identify genomic differences between the two genera that putatively enable them to colonise the distinct niches of topsoil and subsoil environment. Finally, through a gene tree-species tree reconciliation approach the authors provide a comprehensive scenario for the genome evolution of the *Gagatemarkarchaeaceae* family and the important metabolic traits gained in each transition. This, and previous studies from the same group, manage to illustrate that these deeply rooted Thaumarchaeal lineages have quite a broad geographical distribution and consequently should be considered important players in biogeochemical cycling in moderate terrestrial environments.

In general, the study is well written, the results well presented and the figures and data provided very well documented and organised, which will be a great resource for the community.

Importantly, they revisit the topic of optimal growth temperature of the thaumarchaeal ancestor, previously found to be thermophilic according to available data at the time. They find that in contrast with previous reconstructions, the thaumarchaeal LCA was moderately thermophilic (48°C), with thermophilic non-AOA lineages evolving later. This finding raises some questions, and it would be great if it could be discussed further in the context of our current understanding of the related lineages.

Specifically, if a hot spring or in general a thermophilic environment then is ruled out as a habitat for the thaumarchaeal LCA, can the authors propose a scenario for putative habitat transitions for these early ancestors, taking into account the study of Hua et al. 2018? How is this result reconciled with the thermophilic character of Aigarchaeota? Did the authors find any genomic markers for thermophily among the gene losses in the early ancestors, or gained along the evolution of the "thermophilic clade" of nonAOA Thaumarchaeota?

Specific comments:

Supplementary Data 8:

Since the lineages marked in red have no 16S rRNA gene assembled, shouldn't the 6th column title be "Educated guess based on phylogenomic tree topology"?

Supplementary Data 12:

Why only 6 out of the 15 assembled genomes are represented here?

line 134: Aspartate transaminase catalyzes a reversible reaction that functions in both the biosynthesis and degradation of aspartate. Do you find also the second pathway enzyme, aspartate ammonia-lyase (4.3.1.1)?

Some of the genomes also seem to encode the glycine cleavage system, worth mentioning.

line 138: Can the authors add the distribution of the *fadD* gene in Supplementary Data 10, as well as the acquired genes mentioned in paragraph 249-259, so that the readers have an overview?

lines 251-258, 260-266, 268-269, 276, 278-280, 287 : does the phylogeny suggest any potential donors for these families?

Reviewer #2:

Remarks to the Author:

In this manuscript, Sheridan and coauthors explore genomic data of Thaumarchaeota (GTDB: class Nitrososphaeria) associated with a niche separation in soils, with one predominantly enriched in topsoils (0-15 cm) and another enriched in subsoils (30-60 cm). They increase the genome representation in these two groups by adding 15 genomes to the previously published 6. They show that these two groups represent two genera from the same family, and that its phylogenetic placement can inform about the evolution of other thaumarchaea, including ammonia-oxidizing lineages that have received considerable attention. They perform similarity searches of metabolic genes and suggest aerobic respiration may be common in this group. Moreover, they perform evolutionary analyses that show trends of genome expansion in this group by horizontal gene transfer and gene duplication.

These results add to our knowledge of the history and diversity of Thaumarchaeota. For example, the authors obtain conclusions that oppose previous reports of a possible thermophilic origin of ammonia-oxidizing thaumarchaea. Moreover, some of the obtained results may complement previous reports about evolutionary dynamics in the archaeal domain. Yet, all in all, these results have a relatively narrow scope, as they mostly revolve about the ecology and evolution of two closely related genera, even if their niche separation is indeed intriguing.

The analyses are sound, and the results are clearly explained (with only a few exceptions, see below). The methods also seem reproducible, although the generated alignments and phylogenies are still missing.

Due to all the above, I only have a few minor comments:

Metabolic gene searches

L. 65-67. While I doubt these genomes may indeed contain amoA and amoB, it would be good to double-check with more sensitive methods at the protein level. BLASTP, PSIBLAST and HMM searches would be better alternatives to BLASTN to confirm that these genes are missing.

L. 108-177. This section is strongly focused on genes shared by both genera, and contains two paragraphs about the Gagatemaerchaeum genus, but no specific results about the Subgagatemaerchaeum genus. Is there nothing noteworthy in this genus? Even if this is the case, reporting no genus-specific noteworthy metabolic features may be interesting by itself.

Sometimes pathway searches are determined by the presence or absence of a single marker. More in-depth checks can be meaningful. For example:

L. 113-116. Might the rbL subunit be missing in other genomes due to incompleteness? Are all RuBisCO genes a recent acquisition? Checking for the small subunit and other functionally related genes, as well as their synteny conservation (e.g. are they clearly inserted/missing in a specific genomic locus) would add appropriate corroborating evidence and more insights into the nutrition of this family.

Single-gene trees and phylogenetics data

Fig. S3-S5. Did I miss how protein sequences were selected for these protein phylogenies? Please clarify.

Please also provide access to alignments and phylogenies.

OGT prediction

The authors make a strong point in distinguishing their result and that of Abby et al 2020, which they explain by their expanded genome set. However, they also employed different methodology to predict OGT (Tome vs 16S rRNA G-C stem compositions). For example, the Nitrosocaldales appear to be less thermophilic in this study. Could the authors clarify whether the different methodology may have had a strong impact in these conclusions?

Text

L. 49. One might argue a "perfect model" would require additional properties. Maybe soften?

L. 60. "lineages" here is ambiguous, especially considering that GTDB-Tk classifies all genomes within known genera, and that these groups were well sampled at the 16S rRNA level (Fig S2).

L. 63-64. Maybe "represented as" is clearer than "represented by" (to avoid confusing f_UBA183 with a subset of I.1c?

L. 65. It is unclear in this sentence how these two genomes "are related" to these two groups. Could this mean that HMT = f_UBA141? Otherwise, for consistency with the previous line, it would be good to establish what equivalent exists in GTDB for HMT. Could the authors clarify?

L. 77. Since these are likely two genera within the same family and not particularly divergent groups, maybe simply "split" is preferable to "deep split"

L. 79. To facilitate comparisons with the literature (e.g. Abby et al 2020), it would be helpful if the authors could include order-level nomenclature of the AOA genomes.

L. 80-85. Please clarify if the depth of these samples is known at all. The relevance of these observations (higher abundance of one group over the other) is not obvious.

L. 89. Here again, "early diverging" seems unclear. Probably the authors mean early diverging in the context of their position relative to AOA, but it may be worth emphasizing this group for what it is and not mainly on their relative position.

L. 102-107. This paragraph is a bit awkward, with the nomenclature coming out of the blue, and an incomplete sentence ending the paragraph. I recommend to introduce the naming in a more natural way to enhance readability.

L. 120-121. "Enzymes of the GH135 and GT39 are involved (...)". Maybe "Enzymes of the GH135 and GT39 CAZyme families are involved (...)?"

L. 153. Would not "the acid tolerance" be clearer as "acid tolerance"? I trust the authors have a better command of English than I do; I am pointing this out just in case.

L. 202-204. Are these genes therefore present in the common ancestor?

L. 212. If 48 degrees Celsius is considered "moderately thermophilic", I believe the authors should not use a clear "mesophilic" for the AOA LCA, which is estimated to grow optimally at a temperature range between 43 and 46 degrees Celsius (Fig 4). Maybe simply stating the numbers would suffice here. Moreover, the authors state "all AOA (...) form a mesophilic clade", but the Nitrosocaldales remain thermophilic (values unclear but they seem 55-60 or higher in this figure) in this analysis. Please clarify.

L. 239-241. The number of newly acquired families that are duplicated is not clear in Suppl. Data 18.

L. 243. What do the authors mean by “punctuated” in this context, and how is this visible from Suppl. Fig. S9?

L. 234-L.246. It’s unclear why the authors emphasize inter-phyla transfers (L. 234-235 – although it needs to be clarified in the text that Fig. 5 only includes inter-phyla transfers) compared to intra-phylum transfers (L. 246) when it comes to genome expansion of this family. Based on the two provided figures, the impact of each of these is of a similar order.

L. 247. Can the authors provide any kind of reliability assessment (or disclaimer otherwise) of these values? Could phylogenetic artefacts (e.g. LBA, considering the long branch to HTT) cause an overestimate of transfers?

L. 310-311. Given that the authors must have these estimates in the collapsed clade of Fig 5, would it not make sense to show them and mention them here as well, rather than just referring to a previous study?

L. 322-323. Exactly how these different gene loss rates favour one scenario over the other is unclear. Please clarify.

L. 396-397. Using what similarity search software?

L. 438-441. Only assemblies are explicitly mentioned, but no other analysis (other than for result interpretation).

L. 650. Suppl. Text indicates that “+G” may be missing from the evolution model in the legend.

Reviewer #3:

Remarks to the Author:

General comments:

Overall, I found this to be an extremely interesting manuscript documenting the discovery and genomic characterization of yet another novel, deeply-rooted family of Thaumarchaeota. However, I would recommend adding more ecological context to the manuscript. For example, what factors (carbon, saturation, oxygen, temperature, plants, etc.) could be contributing to this large divergence between gaga and subgaga? This is definitely an interesting finding and slightly better framing could make this paper much more interesting to a broader audience. I would also consider bringing up the HTT clade in the abstract; or at least highlight that it's not just this new gagatem family but also a clade sister to the HMT. I would also consider combining the Results + Discussion sections to make it easier on the reader, but that's just my personal preference. I would recommend more description of the importance of the genes that are present/missing. Listing all the genes in certain groups would be more meaningful if it was discussed more immediately why they are of potential interest. Below I have included some more detailed/specific comments to be addressed by the authors.

Specific comments:

If you are trying to prove the absence of a gene (amoA), would 50% complete really be enough?

How reliable is the optimal growth temperature prediction? (particularly since this is the big evolution takeaway).

Line 60: Are these all the archaeal MAGs generated from binning or just the thaumarchaeal MAGs?

Line 75: List Bog-1369 twice

Line 178: TYPO subheading should be "Gagatemarchaeaceae genomic differences between topsoil and subsoil lineages"

Lines 188-194: The comparison of topsoil and subsoil genome content would be better supported with a comparison of completeness. From Table 1, it seems like the subsoil MAGs are slightly more complete and less contaminated.

Line 194: What is the gagatemarchaeum optimal T? Only list sub

Line 196: First line confusing, reword/combine with second sentence

Line 343: Bin_refinement module ADD in metawrap (I know what they were referring to because I use metawrap, but they should state the pipeline here)

Line 366: Why 45% instead of 50% completeness? To keep in one of their low-quality MAGs?

Line 404: Is there precedent for increasing by 19C?

Figure Legends: I would recommend adding more detail to some of these legends, especially which dataset was included.

Figure 1. Misleading not to include other autotrophy pathways, e.g. HP/HB, especially for the AOA. Ammonia oxidation genes?

Figure 2. The A and B legends are switched. The prevalence in peat doesn't seem high enough to warrant the family level namesake.

Figure 3. The colors are switched between top and subsoil from Figure 2. The axes don't span the range.

Figure 5. Inter-phyla? Line 419 says "intra-LGTs" and there is no reference to what other phyla they gained genes from. Maybe add a third color to the gradient to see the differences more clearly.

Figure 6. Numbers in parentheses are not described in the legend. Odd way to order to LCA. Evidence that RuBisCO is gain not loss?

Point-by-point response to the reviewer's comments for "Genomic determinants of soil depth niche partitioning in Gagatemarkarchaeaceae, a novel family of deeply-rooted Thaumarchaeota" to Nature Communications.

REVIEWER COMMENTS

Reviewer #1:

The study of Sheridan et al. focuses on the evolution of deeply rooted non-ammonia oxidizing Thaumarchaeota (currently reclassified as class Nitrososphaeria) and specifically the mechanisms of genomic evolution underpinning the diversification of lineages that colonized terrestrial subsoil and topsoil niches. The authors assemble 15 novel metagenomes from topsoil and subsoil environments, and through phylogenomic analyses propose that the lineage of I.1c Thaumarchaeota be classified as a novel archaeal family, named Ca. Gagatemarkarchaeaceae, comprising two genera (Ca. Gagatemarkarchaeum and Ca. Subgagatemarkarchaeum) and 17 species. Through comparative genomics, they characterise the putative metabolism and identify genomic differences between the two genera that putatively enable them to colonise the distinct niches of topsoil and subsoil environment. Finally, through a gene tree-species tree reconciliation approach the authors provide a comprehensive scenario for the genome evolution of the Gagatemarkarchaeaceae family and the important metabolic traits gained in each transition. This, and previous studies from the same group, manage to illustrate that these deeply rooted Thaumarchaeal lineages have quite a broad geographical distribution and consequently should be considered important players in biogeochemical cycling in moderate terrestrial environments.

In general, the study is well written, the results well presented and the figures and data provided very well documented and organised, which will be a great resource for the community.

Importantly, they revisit the topic of optimal growth temperature of the thaumarchaeal ancestor, previously found to be thermophilic according to available data at the time. They find that in contrast with previous reconstructions, the thaumarchaeal LCA was moderately thermophilic (48°C), with thermophilic non-AOA lineages evolving later. This finding raises some questions, and it would be great if it could be discussed further in the context of our current understanding of the related lineages.

Specifically, if a hot spring or in general a thermophilic environment then is ruled out as a habitat for the thaumarchaeal LCA, can the authors propose a scenario for putative habitat transitions for these early ancestors, taking into account the study of Hua et al. 2018? How is this result reconciled with the thermophilic character of Aigarchaeota?

Reply: We apologise for the presence of a typo in our abstract, ruling out a thermophilic environment for the thaumarchaeal LCA while we meant the AOA LCA (as described in the main text). This has been corrected in the revised version, as follows: "[...] also refuted the previous hypothesis of a thermophilic last common ancestor of the ammonia-oxidising archaea" (L.21-22). This analysis therefore does not contrast with the thermophilic character of Aigarchaeota and as can be seen from Fig 4, there is a trend for higher OGT for earlier ancestors in the evolutionary history of this lineage.

Did the authors find any genomic markers for thermophily among the gene losses in the early ancestors, or gained along the evolution of the "thermophilic clade" of nonAOA Thaumarchaeota?

Reply: We have now further investigated the evolutionary history of the reverse gyrase, rgy, which is considered a hallmark of thermophily in prokaryotes, and have compared our results to Abby et al 2020. Abby et al predicted that rgy was present in the AOA LCA, based on the COUNT tool, which uses presence and absence data for ancestral reconstruction, without incorporation of gene tree

information. In our analysis, we now examine the phylogeny of this gene with a diverse taxonomic sampling. The rgy gene is present in J079 (the only AOA genome in which this gene was detected). Interestingly, the rgy homologs in the thermophilic non-AOA were independently acquired from different donors, rather than vertically inherited from a common ancestor. Therefore, rgy is not predicted to have been present in the AOA LCA (Supp Fig 6). Additional text has been added to describe this (L.237-241).

Specific comments:

Supplementary Data 8:

Since the lineages marked in red have no 16S rRNA gene assembled, shouldn't the 6th column title be "Educated guess based on phylogenomic tree topology"?

Reply: Thanks for spotting this error. This has now been amended.

Supplementary Data 12:

Why only 6 out of the 15 assembled genomes are represented here?

Reply: Several genomes are not represented in this table as no putatively secreted CAZymes were detected in those genomes. This has now been stated more explicitly at the bottom of the table. Please note that previous Supp Data 12 is now Supp Data 13.

line 134: Aspartate transaminase catalyzes a reversible reaction that functions in both the biosynthesis and degradation of aspartate. Do you find also the second pathway enzyme, aspartate ammonia-lyase (4.3.1.1)?

Some of the genomes also seem to encode the glycine cleavage system, worth mentioning.

Reply: We did not detect the aspartate ammonia-lyase (4.3.1.1) for the generation of fumarate from aspartate in the Gagatemarchaeaceae, only the reversible oxaloacetate generating pathway. We have now included the prevalence of aspartate ammonia-lyase across the whole genome set in Supp Data 10. We have also included the presence of the glycine cleavage system in the text, as follows: "They also encode genes for the degradation of [...] glycine (glycine cleavage system) [...]" (L.142-143).

line 138: Can the authors add the distribution of the fadD gene in Supplementary Data 10, as well as the acquired genes mentioned in paragraph 249-259, so that the readers have an overview?

Reply: We have now assessed the distribution of fadD and the other fatty acid degradation genes and included this analysis in Supp Data 10. Additionally, we have included the distribution of iol BCEG, purDHM, kdpABC, pntAB, and fno.

lines 251-258, 260-266, 268-269, 276, 278-280, 287 : does the phylogeny suggest any potential donors for these families?

Reply: Our approach enabled detection of transfer of these genes but can unfortunately not predict the potential donors. Indeed, the gene trees used for reconciliation represent a target group rather than an exhaustive representation of the tree of life, so would not be appropriate for making these kinds of predictions. We would also note that the ancient nature of the acquisitions makes it much more difficult to determine donors than more recent acquisitions. Indeed, in the 100's of millions of years since the acquisition, this gene family could have been transferred to numerous recipients, who themselves have undergone extensive speciation (i.e., the actual donor is unlikely to still exist). For example, if we look at Supplementary Fig. 3 (a single gene phylogeny using genes from an expanded phylogeny of prokaryotic genomes, from both archaea and bacteria), we could reasonably predict that this gene was likely acquired multiple times into the Thaumarchaeota. The donor of acquisition A cannot be determined from this. One might be tempted to say that acquisition B came from an ancient Actinobacteria, but this would be uncertain given the available evidence.

Reviewer #2:

In this manuscript, Sheridan and coauthors explore genomic data of Thaumarchaeota (GTDB: class Nitrososphaeria) associated with a niche separation in soils, with one predominantly enriched in topsoils (0-15 cm) and another enriched in subsoils (30-60 cm). They increase the genome representation in these two groups by adding 15 genomes to the previously published 6. They show that these two groups represent two genera from the same family, and that its phylogenetic placement can inform about the evolution of other thaumarchaea, including ammonia-oxidizing lineages that have received considerable attention. They perform similarity searches of metabolic genes and suggest aerobic respiration may be common in this group. Moreover, they perform evolutionary analyses that show trends of genome expansion in this group by horizontal gene transfer and gene duplication.

These results add to our knowledge of the history and diversity of Thaumarchaeota. For example, the authors obtain conclusions that oppose previous reports of a possible thermophilic origin of ammonia-oxidizing thaumarchaea. Moreover, some of the obtained results may complement previous reports about evolutionary dynamics in the archaeal domain. Yet, all in all, these results have a relatively narrow scope, as they mostly revolve about the ecology and evolution of two closely related genera, even if their niche separation is indeed intriguing.

Reply: We thank the reviewer for their generally positive appraisal of our work. Regarding the scope of this work, while we do indeed focus on two genera of the same family, we propose that niche partitioning in this group of organisms can serve as model for habitat transitions between similar ecosystems (topsoil-subsoil) rather than more drastic transitions as often analysed (soil-marine). These transitions between similar ecosystems are important as they likely represent most transitions. Therefore, this work has wider implications into understanding ecology and evolution of the archaea domain, and we believe this finding is of sufficiently broad interest for the readership of Nature Communications.

The analyses are sound, and the results are clearly explained (with only a few exceptions, see below). The methods also seem reproducible, although the generated alignments and phylogenies are still missing.

Reply: Most relevant phylogenies have been included in the supplementary files. In addition, we have now deposited the alignments and phylogenies in an online repository at <https://github.com/SheridanPO-Lab/I.1c-Group>.

Due to all the above, I only have a few minor comments:

Metabolic gene searches

L. 65-67. While I doubt these genomes may indeed contain *amoA* and *amoB*, it would be good to double-check with more sensitive methods at the protein level. BLASTP, PSIBLAST and HMM searches would be better alternatives to BLASTN to confirm that these genes are missing.

Reply: We have now performed several protein-level based searches, including BLASTp, HMM and GHOSTX searches, and those genes were also not found using those approaches. The text has been amended as follows: "The ammonia monooxygenase *amoA* or *amoB* genes were not detected in any of the 15 genomes using BLASTn¹⁸ or BLASTp against custom databases of *amoA* and *amoB* sequences¹⁹, by GhostKOALA⁸, or by hmmsearch [...]" (L.69-70).

L. 108-177. This section is strongly focused on genes shared by both genera, and contains two

paragraphs about the Gagatemaarchaeum genus, but no specific results about the Subgagatemaarchaeum genus. Is there nothing noteworthy in this genus? Even if this is the case, reporting no genus-specific noteworthy metabolic features may be interesting by itself.

Reply: In this section we discuss cytochrome bd ubiquinol oxidase gene *cydA* which is present in the UBA183 and Fn1, which are in fact Subgagatemaarchaeum. We have now altered the text to make this more explicit “the microaerobic respiration terminal oxidase, cytochrome bd ubiquinol oxidase gene *cydA* was present in the Subgagatemaarchaeum genomes, UBA183 and Fn1” (L.153-155). Another noteworthy aspect of the genus is that its members encode a different type of ATPase than the Gagatemaarchaeum. The text has been edited to make this more explicit “The topsoil lineages (Gagatemaarchaeum) possess the acid-tolerant V-type ATPase and most subsoil lineages (Subgagatemaarchaeum) encode the A-type ATPase” (L.192-193).

Sometimes pathway searches are determined by the presence or absence of a single marker. More in-depth checks can be meaningful. For example:

L. 113-116. Might the *rbcl* subunit be missing in other genomes due to incompleteness? Are all RuBisCO genes a recent acquisition? Checking for the small subunit and other functionally related genes, as well as their synteny conservation (e.g. are they clearly inserted/missing in a specific genomic locus) would add appropriate corroborating evidence and more insights into the nutrition of this family.

Reply: It is possible that the *rbcl* genes may not be detected in some genomes due to incompleteness (the likelihood being proportional to incompleteness), but it is very unlikely that this would explain the absence of these genes in a multi-genome lineage (decreasing in likelihood with the number of genomes in the lineage and their completeness).

Gagatemaarchaeaceae genomes possessed the type III ribulose-bisphosphate carboxylase, which consists only of the large chain subunit (PMID: 12730164). We have altered the text to make this clearer “Only three topsoil genomes possess the type III ribulose-bisphosphate carboxylase (*rbcl*)” (L.118-119).

In terms of functionally related genes, Beam et al 2014 proposed that Thaumarchaeota utilised a ribose 1,5-bisphosphate isomerase [EC:5.3.1.29] in its RuBisCO system and indeed we also see this co-occurring *rbcl* in our expanded set of Thaumarchaeota. This data has been added to Supp Data 10 and text describing this has been added to the manuscript “...and the ribose 1,5-bisphosphate isomerase (predicted to be involved in thaumarchaeotal RuBisCo), indicating carbon fixation through the RuBisCo system (Supplementary Data 10).” (L.118-121).

Additionally, we performed the suggested synteny analysis of the predicted RuBisCO system. This revealed that *rbcl*, the ribose 1,5-bisphosphate isomerase and ribose-phosphate pyrophosphokinase are adjacent to each other on the genome, which strengthens the prediction for a RuBisCO system in these organisms (Supp Data 11). The following text has been added to describe these results “These two genes and the ribose-phosphate pyrophosphokinase were found to be adjacent to each other in these genomes, but all three genes were absent from other members of the family” (L.120-123).

Single-gene trees and phylogenetics data

Fig. S3-S5. Did I miss how protein sequences were selected for these protein phylogenies? Please clarify.

Reply: We have now included detailed descriptions of how the protein sequences for these phylogenies were obtained in the Supplementary Methods section (L.57-83).

Please also provide access to alignments and phylogenies.

Reply: As mentioned above, we have now deposited the alignments and phylogenies in an online repository at <https://github.com/SheridanPO-Lab/I.1c-Group> .

OGT prediction

The authors make a strong point in distinguishing their result and that of Abby et al 2020, which they explain by their expanded genome set. However, they also employed different methodology to predict OGT (Tome vs 16S rRNA G-C stem compositions). For example, the Nitrosocaldales appear to be less thermophilic in this study. Could the authors clarify whether the different methodology may have had a strong impact in these conclusions?

Reply: The Abby et al analysis did not include the mesophilic genomes of the Nitrosocaldales (one reported here, but several more have been assembled from marine habitats since that study was published) and this likely contributes to the differences in our results. In addition to the larger dataset, we also believe that the different methodologies may have impacted these conclusions. Predicting OGT from a large number of different genome-derived features (as performed in Tome) is likely more accurate than based on a single gene (16S rRNA G-C stem compositions). However, we prefer avoiding such conflicting comments in the manuscript, hence the omission of this information, as the larger dataset has likely a greater impact on the results.

Text

L. 49. One might argue a “perfect model” would require additional properties. Maybe soften?

Reply: We agree and have now replaced by “strong model”.

L. 60. “lineages” here is ambiguous, especially considering that GTDB-Tk classifies all genomes within known genera, and that these groups were well sampled at the 16S rRNA level (Fig S2).

Reply: We have replaced the word “lineages” with “species”, as their novelty as species genome representatives could be shown by GTDB-Tk, AAI and 16S rRNA similarity (SD1-3)” (L.62).

L. 63-64. Maybe “represented as” is clearer than “represented by” (to avoid confusing f_UBA183 with a subset of I.1c?

Reply: Thank from this suggestion. We have made this change.

L. 65. It is unclear in this sentence how these two genomes “are related” to these two groups. Could this mean that HMT = f_UBA141? Otherwise, for consistency with the previous line, it would be good to establish what equivalent exists in GTDB for HMT. Could the authors clarify?

Reply: Modified as follows to clarify “Two genomes classified as members of the uncharacterised f_UBA141 family, a family closely related to the heterotrophic marine Thaumarchaeota (HMT)^{13,14} (classified as f_UBA57 in GTDB) “ (L.65-67).

L. 77. Since these are likely two genera within the same family and not particularly divergent groups, maybe simply “split” is preferable to “deep split”

Reply: we have replaced “deep” by “significant” (L.80).

L. 79. To facilitate comparisons with the literature (e.g. Abby et al 2020), it would be helpful if the authors could include order-level nomenclature of the AOA genomes.

Reply: No AOA genomes are mentioned on this line, so we presume the comment is for Figure 1. As figure 1 is already quite dense and the focus of the study is on the Gagatemarchaeaceae not the AOA, we prefer not to add the order-level of the AOA genomes in it, as the order-level taxonomy is already indicated in Supp Data 1.

L. 80-85. Please clarify if the depth of these samples is known at all. The relevance of these observations (higher abundance of one group over the other) is not obvious.

Reply: In this paragraph, one sequence from each of the two groups was used as a representative of top- and sub-soil Group I.1c, respectively. These two 16S rRNA gene sequences were compared to public 16S rRNA sequencing libraries, providing hits for more than 67,000 soils. As such, we could not manually check the information regarding soil depth in so many entries, and soil depth is not a mandatory field in NCBI entry. We clarified this in the text as follows “Using representative 16S rRNA gene sequences from each of the two Group I.1c lineages, it was observed that...” (L.83-85).

L. 89. Here again, “early diverging” seems unclear. Probably the authors mean early diverging in the context of their position relative to AOA, but it may be worth emphasizing this group for what it is and not mainly on their relative position.

Reply: We removed “early” to avoid confusion.

L. 102-107. This paragraph is a bit awkward, with the nomenclature coming out of the blue, and an incomplete sentence ending the paragraph. I recommend to introduce the naming in a more natural way to enhance readability.

Reply: We have now modified the paragraph by introducing the taxonomy as follows: “With regards to formal taxonomic classification, we selected the genome bog-1369 as ...” (L.105). The last sentence has also been rephrased as follows: “Full classification notes are detailed in the supplementary text.” (L.109-110).

L. 120-121. “Enzymes of the GH135 and GT39 are involved (...)”. Maybe “Enzymes of the GH135 and GT39 CAZyme families are involved (...)”?

Reply: modified as suggested (L.130).

L. 153. Would not “the acid tolerance” be clearer as “acid tolerance”? I trust the authors have a better command of English than I do; I am pointing this out just in case.

Reply: Thanks for pointing out. This has now been amended. (L.162).

L. 202-204. Are these genes therefore present in the common ancestor?

Reply: We would suggest that this is not the case, but we would be very reluctant to make that particular prediction at this time, as each of these gene families possess only 1-2 representative sequences from the HTT clade.

L. 212. If 48 degrees Celsius is considered “moderately thermophilic”, I believe the authors should not use a clear “mesophilic” for the AOA LCA, which is estimated to grow optimally at a temperature range between 43 and 46 degrees Celsius (Fig 4). Maybe simply stating the numbers would suffice here. Moreover, the authors state “all AOA (...) form a mesophilic clade”, but the Nitrosocaldales remain thermophilic (values unclear but they seem 55-60 or higher in this figure) in this analysis. Please clarify.

Reply: We have now reworded the text as follows: “Ridge regression of extant genome optimal growth temperatures (OGTs) across the thaumarchaeotal species tree indicates that the thaumarchaeotal LCA had an OGT of 48°C, with a gradual reduction in OGT to 43°C for the AOA LCA (Fig. 4). Our analysis predicts that the AOA and multiple lineages of non-AOA Thaumarchaeota form a mesophilic clade, except for some thermophilic genomes belonging to the Nitrosocaldales lineage. The non-AOA Thaumarchaeota lineage encompassing the Dragon (DS1, UBA164 and UBA160), Beowulf (BS3 and BS4) and Conexivisphaera calida NAS-02 genomes is sister to this mesophilic clade.” (L.225-231).

L. 239-241. The number of newly acquired families that are duplicated is not clear in Suppl. Data 18.

Reply: This number is given in the “# Gene families with > 1 copy” column. The word “duplicated” has been added in parenthesis to make this clearer. Please note that previous Supp Data 18 is now Supp Data 19.

L. 243. What do the authors mean by “punctuated” in this context, and how is this visible from Suppl. Fig. S9?

Reply: The punctuation score is estimated by the formula indicated in blue on the figure. As indicated in the legend, it represents the sum of events in the 10% of branches with the highest event numbers divided by 10% of the sum of events into all branches ($\sum \text{events in top 10\%} / (\sum \text{events in all branches} * 0.1)$). This score was estimated in a previous published study (doi: 10.1038/s41467-022-31847-7). We have added the text “(i.e., events were less concentrated in a small number of species tree branches, as indicated by a lower punctuation score²)” (L.273-274) to explain the term “punctuated” in this context and reference the original paper describing the punctuation score.

L. 234-L.246. It’s unclear why the authors emphasize inter-phyla transfers (L. 234-235 – although it needs to be clarified in the text that Fig. 5 only includes inter-phyla transfers) compared to intra-phylum transfers (L. 246) when it comes to genome expansion of this family. Based on the two provided figures, the impact of each of these is of a similar order.

Reply: Separating the inter- and intra-phyla events is indeed not important for the genome expansion itself. However, emphasizing inter-phyla transfers is critical for novel metabolic acquisition, potentially enabling transition into terrestrial environments. A sentence has been added to clarify this point “...genome expansion occurred during the transition into terrestrial environments (Supplementary Fig. 8). Genome expansion was likely initiated by numerous intra- and inter-phyla gene transfers, with the latter being crucial for providing novel metabolic acquisition, enabling environmental transition. Two periods of extensive acquisition...”. (L.256-258). We also modified the legend title as follows: “Acquisition of novel gene families (through inter-phyla transfers) and ...” (L.735).

L. 247. Can the authors provide any kind of reliability assessment (or disclaimer otherwise) of these values? Could phylogenetic artefacts (e.g. LBA, considering the long branch to HTT) cause an overestimate of transfers?

Reply: As the inferences from gene tree-species tree reconciliation make use of information from the gene tree topology, there is the potential (as with other phylogenetic methods) for the number of transfers to be inflated by inference error or, potentially, other sources of gene tree-species tree discordance such as incomplete lineage sorting. In the revision, we have mentioned this caveat and also unpacked the meaning of these values in the text (L.261-268).

L. 310-311. Given that the authors must have these estimates in the collapsed clade of Fig 5, would it not make sense to show them and mention them here as well, rather than just referring to a previous study?

Reply: The cited study is a targeted investigation of the AOA Nitrososphaerales, rather the I.1c Group targeted study presented here, and thus contains a much more diverse AOA dataset than the one used for Figure 5, so the estimates from the cited paper remain more accurate for the Nitrososphaerales.

L. 322-323. Exactly how these different gene loss rates favour one scenario over the other is unclear. Please clarify.

Reply: The reasoning is that genes present in the family's LCA were likely beneficial to the LCA in its native habitat. A greater loss of these ancestral genes would indicate a greater habitat change between the family LCA and Subgagamarchaeum (which inhabits subsoil), than in Gagatemarchaeum (which inhabits topsoil) -though of course the evidence is indirect. We have expanded on this point in the text to clarify the reasoning (L.354-355).

L. 396-397. Using what similarity search software?

Reply: We used Diamond BLASTp as similarity search software. This has now been included in the text "arCOG database, was estimated using Diamond BLASTp". (L.431).

L. 438-441. Only assemblies are explicitly mentioned, but no other analysis (other than for result interpretation).

Reply: This has been amended in the text "P.O.S. assembled the 15 new genomes and performed genomic analyses."(L.473-474).

L. 650. Suppl. Text indicates that "+G" may be missing from the evolution model in the legend.

Reply: Thanks for noting this omission. This has now been corrected. (L.697).

Reviewer #3 (Remarks to the Author):

General comments:

Overall, I found this to be an extremely interesting manuscript documenting the discovery and genomic characterization of yet another novel, deeply-rooted family of Thaumarchaeota. However, I would recommend adding more ecological context to the manuscript. For example, what factors (carbon, saturation, oxygen, temperature, plants, etc.) could be contributing to this large divergence between gaga and subgaga? This is definitely an interesting finding and slightly better framing could make this paper much more interesting to a broader audience.

Reply: This is a very interesting comment, which we thought a lot about before submitting. Unfortunately, we do not have biogeochemical data or extensive soil chemical or physical analyses performed at the time of sampling. At the time of submission, the soils had been stored for >2 years at 4°C, so we felt that these data would misrepresent the ecosystem. Therefore, we preferred to avoid discussing the possible factors possibly contributing to this large divergence between Gagatemarchaeum and Subgagamarchaeum as it could be perceived as highly speculative.

I would also consider bringing up the HTT clade in the abstract; or at least highlight that it's not just this new gagatem family but also a clade sister to the HMT.

Reply: We agree this is an important point and have now included HTT clade in the abstract as follows: "Here, 15 new deeply-rooted thaumarchaeotal genomes were assembled from acidic topsoils (0-15cm) and subsoils (30-60 cm), corresponding to two genera of terrestrially prevalent Gagatemarchaeaceae (previously known as thaumarchaeotal Group I.1c) and to a novel genus of heterotrophic terrestrial Thaumarchaeota. " (L.13-15).

I would also consider combining the Results + Discussion sections to make it easier on the reader, but that's just my personal preference.

Reply: Thank you for this suggestion. On balance, and considering the comments of the other reviewers, we decided to keep a separate Results and Discussion section.

I would recommend more description of the importance of the genes that are present/missing. Listing all the genes in certain groups would be more meaningful if it was discussed more immediately why they are of potential interest.

Reply: We have endeavoured throughout this paper to ensure that the importance of the genes are discussed immediately alongside their mention, however after a careful revision of the text we have identified several places where descriptions could be improved. Text has been amended as follows:

PQQ-dependent dehydrogenases: **L.175-178**

“Pyrroloquinoline quinone (PQQ)-dependent dehydrogenases catalyse the oxidation of a variety of alcohols and sugars. These genes were highly expressed in marine environments and predicted to play an important physiological role in the heterotrophic marine Thaumarchaeota (HMT)13, 14. PQQ-dependent dehydrogenases are also present in most Gagatemarchaeum (Fig. 1), with up to 8 genes per genome.”

CAZymes and peptidases: **L.199-202**

“Gagatemarchaeum genomes also possess significantly more CAZymes (involved in carbohydrate degradation) ($P < 0.02$) and peptidases (involved in peptide degradation) ($P < 0.02$) than Subgagatemarchaeum genomes (Fig. 3).”

RuBisCo: **L.319**

“its LCA acquired the ribulose-bisphosphate carboxylase large chain, rbcL, which was shown previously to classify as a Type III RuBisCO11 (involved in carbon fixation) (Fig. 6)”

Below I have included some more detailed/specific comments to be addressed by the authors.

Specific comments:

If you are trying to prove the absence of a gene (amoA), would 50% complete really be enough?

Reply: While genome incompleteness might possibly explain the absence of amoA in some individual genome sequences (proportional to the level of incompleteness), we deem it very unlikely to explain the absence of this gene in whole lineages containing multiple genomes, as the probability of failing to sequence or assemble the gene in any genome drops substantially with each additional sampled genome from the clade.

How reliable is the optimal growth temperature prediction? (particularly since this is the big evolution takeaway).

Reply: TOME utilises a machine learning model, which is based on a trained dataset. It has been used in several prokaryotic systems (see paper’s citations) and is perhaps the most accurate current method, because it uses information from the entire proteome rather than a small number of genes, though – as with all methods – accuracy varies across the tree of life (Sauer and Wang 2019). In the case of Thaumarchaeota, there is very good agreement between experimentally-determined (culture-based) optimal growth temperatures and the TOME predictions (see table S16 from Sheridan et al, 2020, Nat Com, <https://doi.org/10.1038/s41467-020-19132-x>) for AOA mesophiles. This is not the case for AOA thermophiles, explaining why we include an analysis (in parenthesis in Fig4) in which all genome predictions of OGT greater than 45 °C were increased by 19°C, which was the greatest difference observed between empirical and predictions of OGT. Both approaches (with and without adjustment) provided similar conclusions.

The approach with hyperthermophile adjustment was performed to counter any suggestion that our results are caused by underestimations of thermophile OGTs. With the exaggeratedly high OGT predictions, we observe that the OGT of AOA LCA is 46°C, as opposed to the 76°C proposed by Abby et al 2020.

Line 60: Are these all the archaeal MAGs generated from binning or just the thaumarchaeal MAGs?

Reply: We assembled more than 2,000 MAGs and the 15 reported represent the thaumarchaeal MAGs of interest. We modified the sentence as follows “Fifteen Thaumarchaeota metagenome-assembled genomes (MAGs)...” (L.61).

Line 75: List Bog-1369 twice

Reply: Thanks for noting this typo. One of the two “Bog-1369” should be “Bog-1367”. (L.78).

Line 178: TYPO subheading should be “Gagatamarchaeaceae genomic differences between topsoil and subsoil lineages”

Reply: Thanks for noting this typo. One “topsoil” has been replaced by “subsoil”. (L.189).

Lines 188-194: The comparison of topsoil and subsoil genome content would be better supported with a comparison of completeness. From Table 1, it seems like the subsoil MAGs are slightly more complete and less contaminated.

Reply: Differences in completeness and contamination between the two lineages are not statistically significant. In fact, in this comparison we did adjust for genome incompleteness, but omitted to mention this in the text. This oversight has now been corrected as follows:

“Values for CAZymes, peptidases and genome size in this comparison have been adjusted by the genome incompleteness.” (L.207-208).

Line 194: What is the gagatamarchaeum optimal T? Only list sub

Reply: The number indicated reflects the difference between the two groups. To clarify this we have included the average OGTs of both groups in the text, as follows: “the predicted optimal growth temperature of the Subgagatamarchaeum (average 40.5 °C) was slightly higher than that of Gagatamarchaeum (average 36 °C) ($P < 1e-5$) (Fig. 3).” (L.205-207).

Line 196: First line confusing, reword/combine with second sentence

Reply: The first sentence has now been reworded and combined with the second sentence as follows: “Two newly acquired genomes, representing a novel terrestrial genus related to the HMT and the uncharacterised f_UBA141 family (Fig. 1), lack the gene markers for autotrophic carbon fixation (Supplementary Data 10) and possess genes for carbohydrate, peptide, and fatty acid utilisation (Fig. 1).” (L.210-213).

Line 343: Bin_refinement module ADD in metawrap (I know what they were referring to because I use metawrap, but they should state the pipeline here)

Reply: This has now been added: “the Bin_refinement module from MetaWRAP” (L.376-377).

Line 366: Why 45% instead of 50% completeness? To keep in one of their low-quality MAGs?

Reply: The threshold 45% was chosen based on two previous studies (Sheridan et al., 2020, Nat Com and Sheridan et al., 2022, Nat Com) as it provides an adequate threshold between genome completeness and number of genomes for evolutionary analyses.

Line 404: Is there precedent for increasing by 19C?

Reply: We address this in the reply above; briefly, the aim here was to apply a conservative correction in cases where the method used might have underestimated OGT.

Figure Legends: I would recommend adding more detail to some of these legends, especially which dataset was included.

The datasets used in the inference of species trees have now been explicitly stated in Fig. 1, 4 and 5. Additionally, extra text has been added to Fig legend 5 to guide the readers to the Methods section detailing the gene tree-species tree analysis, as follows: “across the Thaumarchaeota phylum tree (Dataset 3) using a gene tree-species tree reconciliation approach as described in the Methods section “Predicting gene content changes across evolutionary history””. (L.738-739).

We have also included text in Fig.6 legends to explain the numbers in parenthesis, based on this and the previous reviewer’s comment: “The numbers in parenthesis correspond to the number of gene families of the named function gained in each ancestor” (L.745-746).

Figure 1. Misleading not to include other autotrophy pathways, e.g. HP/HB, especially for the AOA. Ammonia oxidation genes?

Reply: HP/HB and ammonia oxidation genes have now been added to the figure

Figure 2. The A and B legends are switched. The prevalence in peat doesn’t seem high enough to warrant the family level namesake.

Reply: Thanks for noting this legend inversion. We agree that Gagatemarchaeaceae are present in a large range of environments. However, the higher prevalence was observed in peat soils, and previous research on this lineage was mostly performed in peat soils (e.g. Jurgens et al., 1997, Appl Environ Microbiol; Bomberg et al., 2007, Microb Ecol ; Weber et al., 2015, FEMS Microbiol Ecol; Lin et al., 2015, ISME; Biggs-Weber et al., 2020, Soil Biol Biochem). Therefore, we would prefer to keep this family level name.

Figure 3. The colors are switched between top and subsoil from Figure 2. The axes don’t span the range.

Reply: Thanks for noting this colour inversion. The colours have now been amended in Figure 2 to correspond with the rest for the figures. The axes of Figure 3 have also been expanded.

Figure 5. Inter-phyla? Line 419 says “intra-LGTs” and there is no reference to what other phyla they gained genes from.

Reply: The left panel on Figure 5 indeed represent the inter-phyla events and we have modified the legend title as follows to clarify this: “Acquisition of novel gene families (through inter-phyla transfers) and ...”.

In the text, quantification of the four types of events was indeed estimated with the originations representing the inter-phyla transfers. This has been added to clarify: “... this approach allowed inferring the numbers of duplications, intra-LGTs, losses and originations (inter-LGTs) on each branch...” (L.453).

From our analysis, we cannot clearly determine what the inter-phyla gene donors were. This is very difficult to accurately predict as the acquisitions are ancient and thus the donor likely no longer exists and there has also been 100’s millions of years of these genes being transfer from one to another (i.e., non-vertical acquisition).

Maybe add a third color to the gradient to see the differences more clearly.

We assume that the reviewer is referring to Fig 4 here. A gradient from blue to red seems sensible for indication cold to hot. We would rather not change this, as it may make the figure less easily interpretable.

Figure 6. Numbers in parentheses are not described in the legend. Odd way to order to LCA. Evidence that RuBisCO is gain not loss?

Reply: The legend of the numbers in parentheses have now been added to the legend: “The numbers in parenthesis correspond to the number of gene families of the named function gained in each ancestor.” (L.745-746). We numbered the LCA based on their relative importance for our study, with the focus being on Gagatemarchaeaceae LCA and its sub-lineages.

The prediction that RuBisCO is gained in the “thermophilic” clade comes from its predicted absence in the Thaumarchaeota LCA by gene tree-species tree reconciliation and its presence in the “thermophilic” clade LCA.

Reviewers' Comments:

Reviewer #1:

Remarks to the Author:

This is a resubmission of a study focusing on the mechanisms of genome evolution underpinning the diversification of non-ammonia oxidizing lineages within Thaumarchaea (previously known as I.1c), specifically two lineages (representing two distinct genera) that have adapted to terrestrial subsoil and topsoil niches. 15 novel metagenomes are assembled in this study, which allow for a comprehensive investigation of the metabolic potential of these clades and indicate their involvement in carbon cycling in soils. This, and previous studies from the same group, manage to illustrate that these deeply rooted Thaumarchaeal lineages have quite a broad geographical distribution and consequently should be considered important players in biogeochemical cycling in terrestrial environments.

The authors use new methodologies and their novel extended dataset of non-AOA Thaumarchaea to revisit the issue of the optimal growth temperature of the thaumarchaeal ancestor, previously found to be thermophilic. Their results indicate that the ancestor of Thaumarchaea and AOA was rather mesophilic, with some lineages exhibiting secondary adaptations to thermophily. They explain the methodology and the differences with previous approaches clearly and convincingly, and are aware of its limitations. The authors have addressed all previous comments in a satisfactory manner.

Minor comment:

Fig 2B: The colour codes for Subsoil/Topsoil seem to be reversed in the bar plot. The text states that "Subsoil Group I.1c are twice as prevalent than topsoil Group I.1c in peat (11 versus 6%), whereas topsoil Group I.1c are 4-fold more prevalent than subsoil Group I.1c in more than 67,000 soils (7 versus 2%)", the opposite of what the bars in the figure show.

Reviewer #2:

Remarks to the Author:

I commend the authors for their improvements to the manuscript. It reads better, and all the minor issues I had have been adequately solved or argued. I believe the manuscript is ready for publication.

Reviewer #3:

Remarks to the Author:

The authors have adequately addressed my comments and questions in this revised version of the manuscript.

REVIEWERS' COMMENTS

Reviewer #1 (Remarks to the Author):

This is a resubmission of a study focusing on the mechanisms of genome evolution underpinning the diversification of non-ammonia oxidizing lineages within Thaumarchaea (previously known as I.1c), specifically two lineages (representing two distinct genera) that have adapted to terrestrial subsoil and topsoil niches. 15 novel metagenomes are assembled in this study, which allow for a comprehensive investigation of the metabolic potential of these clades and indicate their involvement in carbon cycling in soils. This, and previous studies from the same group, manage to illustrate that these deeply rooted Thaumarchaeal lineages have quite a broad geographical distribution and consequently should be considered important players in biogeochemical cycling in terrestrial environments.

The authors use new methodologies and their novel extended dataset of non-AOA Thaumarchaea to revisit the issue of the optimal growth temperature of the thaumarchaeal ancestor, previously found to be thermophilic. Their results indicate that the ancestor of Thaumarchaea and AOA was rather mesophilic, with some lineages exhibiting secondary adaptations to thermophily. They explain the methodology and the differences with previous approaches clearly and convincingly, and are aware of its limitations. The authors have addressed all previous comments in a satisfactory manner.

Minor comment:

Fig 2B: The colour codes for Subsoil/Topsoil seem to be reversed in the bar plot. The text states that "Subsoil Group I.1c are twice as prevalent than topsoil Group I.1c in peat (11 versus 6%), whereas topsoil Group I.1c are 4-fold more prevalent than subsoil Group I.1c in more than 67,000 soils (7 versus 2%)", the opposite of what the bars in the figure show.

Thank you for pointing out that mistake. The figure has now been corrected.